

# What drives the spatial variability of primary productivity and matter fluxes in the North-West African upwelling system ? A modelling approach and box analysis

Pierre-Amaël Auger[1,2,3], Thomas Gorgues[1], Eric Machu[1], Olivier Aumont[1], Patrice Brehmer[2]

[1]Laboratoire d'Océanographie Physique et Spatiale (LOPS), UMR-CNRS 6523/IFREMER/IRD/UBO, BP70, 29280 Plouzané, France
[2]Laboratoire de l'Environnement Marin (LEMAR), UMR-CNRS 6539/IRD/UBO, place N. Copernic, 29280 Plouzané, France
[3]Instituto Milenio de Oceanografia and Escuela de Ciencias del Mar, Pontificia Universidad Catolica de Valparaiso, Valparaiso, Chile

*Correspondence to*: Pierre-Amaël Auger (pierreamael.auger@gmail.com)

**Abstract.** A comparative box analysis based on a multi-decadal physical-biogeochemical hindcast simulation (1980–2009) was conducted to characterize the drivers of the spatial distribution of phytoplankton biomass and production in the North-West (NW) African upwelling system. Alongshore geostrophic flow related to large scale circulation patterns associated with the influence of coastal topography are suggested to modulate the coastal divergence, and then the response of nutrient upwelling to wind forcings. In our simulation, this translates into a coastal upwelling of nitrate being significant in all regions but the Cape Blanc (CB) area. However, upwelling is found to be the dominant supplier of nitrate only in the northern Saharan Bank (NSB) and the Senegalo-Mauritanian (SM) regions. Elsewhere, nitrate supply is dominated by meridional advection, especially off Cape Blanc. Phytoplankton displays a similar behaviour with a supply by lateral advection which equals the net coastal phytoplankton growth in all coastal regions except the Senegalo-Mauritanian area. Noticeably, in the Cape Blanc area, the net coastal phytoplankton growth is mostly sustained by high levels of regenerated production exceeding new production by more than two fold which is in agreement with the locally weak input of nitrate by coastal upwelling. Further offshore, the distribution of nutrients and phytoplankton is explained by the coastal circulation. Indeed, in the northern part of our domain (i.e. Saharan Bank), the coastal circulation is mainly alongshore resulting in low offshore lateral advection of nutrients and phytoplankton. On the contrary, lateral advection transport coastal nutrients and phytoplankton towards offshore areas in the latitudinal band off the Senegalo-Mauritanian region. Moreover, this latter offshore region benefits from transient southern intrusions of nutrient-rich waters from the Guinean upwelling.





# 1 Introduction

Among the four major eastern boundary upwelling systems (EBUS), the North-West (NW) African upwelling region is the most spatially and seasonally variable one in terms of primary productivity (Carr and Kearns, 2003). This variability may impact the distribution and abundance of fish populations, and their associated fisheries, on a large range of time scales

(Arístegui et al., 2009). It also constrains the dynamics of nutrient and organic carbon export from the coastal margin (Helmke et al., 2005; Muller-Karger et al., 2005). Various studies have established that upwelling-driven nutrient supply is the key factor regulating chlorophyll concentration and primary production off NW Africa (Lathuilière et al., 2008; Messié and Chavez, 2014; Ohde and Siegel, 2010; Pradhan et al., 2006). However, the mechanisms that control the spatio-temporal variability of primary productivity are still poorly understood.

In EBUS, primary production and phytoplankton biomass are first enhanced by the wind-driven coastal upwelling of nutrient-rich waters into the euphotic zone (Allen, 1973). Upwelled waters are redistributed by advection processes, while turbulent mixing accounts for their dilution into surrounding waters. Mixing naturally acts against the build-up of plankton biomass. Indeed, the nutrient utilization can only be optimized by retentive physical mechanisms in the coastal area which enhance microbial remineralization of particulate organic matter and zooplankton excretion, and then regenerated production

through ammonium consumption.

During the last decade, several studies focused on the characterization of the variability of satellite-derived surface chlorophyll in the NW African region which they interpreted in regards of environmental forcings (Lathuilière et al., 2008; Pradhan et al., 2006; Thomas et al., 2001). One of there main findings was that the seasonal variability of wind-forcing is the main driver of the surface chlorophyll seasonal variability. If this main result is expected, it does not give the full account of

20 the latitudinal discrepancies of the seasonal variability of the surface chlorophyll and the related underlying processes within the NW African upwelling system. Indeed, the region between 24° N and 33° N (most of the Moroccan sub-region including the northern Saharan Bank) is characterized by a weak seasonality and chlorophyll confined to the coast. The Cape Blanc area (19–24° N, including the southern Saharan Bank) also presents a weak seasonality but is the site of a persistent offshore extension of the coastal chlorophyll pattern. In the Senegalo-Mauritanian region (10–19° N), chlorophyll is enhanced

together with a large offshore extension during winter/spring, followed by an abrupt drop during summer. Lathuilière et al. (2008) suggest that nutrient limitation is the key factor that explains the weak offshore extension of chlorophyll in the north. In the south, they partly attribute the drop of chlorophyll to the seasonal intrusion of nutrient-depleted waters from the North Equatorial Counter Current (NECC).

Here we propose a modelling approach to gain the first mechanistic understanding of the underlying processes controlling

the spatial variability of primary productivity and to test the satellite-based hypothesis proposed by Lathuilière et al. (2008). In our study, outputs of a multi-decadal physical-biogeochemical hindcast simulation (1980–2009; see Auger et al., 2015) were used to characterize spatially the drivers of phytoplankton biomass and production in the NW African upwelling system



with a particular focus on the mechanisms that control the sensitivity of primary productivity to the wind forcing and the coastal upwelling. To this end, comparative box analysis representing homogeneous sub-regions in the NW African upwelling system has been conducted. Those sub-regions have been defined using the near-surface horizontal circulation patterns. In each box, we analysed the dynamics of primary productivity and nutrients in regards of advective and diffusive matter fluxes at the boundaries and local biological production/uptake. The nature and variability of the matter exported from the coastal margin to the adjacent open ocean were also subsequently depicted.

First, the model configuration is presented with a validation of near-surface circulation and surface chlorophyll biomass using in-situ and satellite data (Section 2.1 and 2.2, respectively). The following section is focused on the description of the meridional variability of annual wind forcings, ocean response and primary productivity as simulated by the model in the different coastal boxes (Section 3.1) and offshore boxes (Section 3.2). These two sections are split in three parts which describe the meridional variability of (i) the wind forcings, current velocity and nitrate fluxes, (ii) the primary production (PP), phytoplankton biomass and phytoplankton fluxes, and (iii) the sources and sinks of nitrate concentration and phytoplankton biomass.

The implication of the model results are discussed in Section 4. In this last section: (i) we explicit the mechanisms driving the sensitivity of coastal upwelling to the wind forcing along the NW African coast (Section 4.1), (ii) we depict the mechanisms driving the meridional variability of coastal phytoplankton biomass and PP (new and regenerated production) in relation with matter transfers (Section 4.2) and (iii) we provide new insights on the processes driving the meridional variability of the offshore extension of the coastal chlorophyll pattern off NW Africa (Section 4.3). First, the Ekman-based relation between upwelling-favourable winds and upwelling intensity, widely used to estimate vertical nutrient inputs in upwelling systems, is hardly questioned by our model results. Second, the weak offshore extension of satellite-based surface chlorophyll north of Cape Blanc is not attributed to nutrient limitation challenging the main hypothesis of Lathuilière et al. (2008) but rather to horizontal advection.

## 2 Material and methods

A multi-decadal hindcast simulation of the physical-biogeochemical dynamics in the NW African upwelling system was run over the period 1980–2009. We used the three-dimensional (3D) primitive equations, sigma-coordinates, free surface Regional Oceanic Modeling System (ROMS – (Shchepetkin and McWilliams, 2005) configured for the NW African upwelling system (Machu et al., 2009; Marchesiello and Estrade, 2009). Model parameterizations, including a parameterization of the Mediterranean outflow, are described by Marchesiello and Estrade (2009). The physical model was coupled to a biogeochemical model (PISCES – Aumont et al., 2003; Aumont and Bopp, 2006) which simulates plankton productivity and carbon biomass based upon the main nutrients (nitrate, ammonium, phosphate, silicate and iron). This model includes two size classes of phytoplankton (nanoflagellates and diatoms), zooplankton (ciliates and copepods) and two classes of detritus (the latter differ by their sinking velocity: 5/30 m day$^{-1}$ for small/large particulate material,



respectively). Phytoplankton growth depends on light, temperature and the external availability in nutrients. Diatoms differ from nanoflagellates by their silicate requirement, higher requirement in iron (Sunda and Huntsman, 1997) and higher half-saturation constant due to larger size. Microzooplankton differs from mesozooplankton by food diet (related to the prey/predator size ratio), grazing rates and mortality parameterization. PISCES has previously been used in global (e.g. Aumont et al., 2003; Aumont and Bopp, 2006; Gorgues et al., 2010), basin-scale (e.g. Gorgues et al., 2005; José et al., 2014) and regional upwelling studies (e.g. Albert et al., 2010; Auger et al., 2015; Echevin et al., 2008).

Heat, solar and water fluxes from the CFSR atmospheric reanalysis (1/3° resolution, NCEP Climate Forecast System Reanalysis, Saha et al., 2010) were used to force our interannual simulation at a 6-hour time scale. Lateral open boundary conditions of both physical and biogeochemical fields were provided by a 5 days archived NEMO-PISCES simulation of the North-Atlantic basin (1/4° resolution, T. Gorgues, pers. comm.). Surface nutrient fertilization were solely provided by iron dust deposition, parameterized using a modelled climatology of the atmospheric dust deposition (for details, see Aumont et al., 2008).

The topography was based on GEBCO 1' resolution (General Bathymetric Chart of the Oceans, http://www.gebco.net). A "child" grid focused on the NW African upwelling (10–35° N / 9–23° W, 1/12° resolution, eddy-resolving) was embedded in a lower resolution "parent" grid (5–40° N / 5–30° W, 1/4° resolution) through a two-way coupling (AGRIF – Debreu et al., 2012). The use of this technique limits the influence of discontinuities emerging from low spatio-temporal resolution of open boundary conditions on the "child" solution, and also produces upscaling effects on the "parent" solution. More details on the simulation can be found in Auger et al. (2015).

### 2.1 Physical-biogeochemical model

### 2.2 Model validation

As previously described by Auger et al. (2015), the general distribution of sea surface temperature (SST) agrees well with AVHRR satellite data although a warm/cold bias of about 1°C exists in offshore/nearshore SST, respectively. The coastal region of cold surface waters is notably thinner in the model particularly off Cape Blanc and, during the upwelling winter season, off Mauritania (see Fig. 2 in Auger et al., 2015). However, the general circulation and its seasonal variability are well reproduced by the model (Fig. 1, winter and summer averages) as attested by climatology of near-surface currents from the model (15m) and derived from satellite-tracked near-surface drifting buoy (1979–present, 1/2° resolution, Lumpkin and Johnson, 2013).

As part of the eastern branch of the North-Atlantic subtropical gyre, the Canary Current flows equatorward along the NW African coast and separates from the coast around Cape Blanc (19–21° N) where it feeds the North Equatorial Current. South of 19° N, a large cyclonic recirculation is found between the south-westward flowing Canary Current and the coast, especially in summer when trade winds extend farther north (see Barton et al., 1998; Mittelstaedt, 1983, 1991). The southern branch of the recirculation gyre is fed by the eastward flowing North Equatorial Counter Current (NECC) which is located



near 10° N in summer and 5° N in winter (Mittelstaedt, 1991; Stramma et al., 2005). Maximum velocity is found equatorward in the coastal upwelling jet (Canary Upwelling Current) north of Cape Blanc where upwelling occurs all year round. The upwelling filaments off Cape Ghir and Cape Boujdour are responsible for strong seaward deflections of the coastal current. As a matter of fact, strong westward velocities off Cape Boujdour most likely limits the drifter sampling over
the Saharan Bank. Surface currents then turn west in the inter-gyre region off Cape Blanc feeding the North Equatorial Current (see Fig. 1). In the Senegalo-Mauritanian region, surface currents are directed south-westward during the winter upwelling season. Alternatively, a moderate poleward current (which can be seen as an extension of the NECC, see Fig. 1) lays south of Cape Blanc both in the model and in the data during summer when upwelling-favourable winds are weak. However, this poleward current does not persist during winter in our simulation as well as in the drifter climatology offshore
of Mauritania. The seasonality of the coastal current in the same latitudinal band, a crucial feature, is nevertheless well simulated with strong equatorward advection in winter and moderate poleward advection in summer. Noticeably, the flow over the slope is always poleward (not shown).

The spatial and seasonal variability of surface chlorophyll is consistent with SeaWiFS satellite data (Fig. 1). In both model and satellite data, chlorophyll concentrations are globally maximum in the coastal upwelling (5–10 mgChl m$^{-3}$) and decrease
offshore toward the subtropical gyre (0.1–0.2 mgChl m$^{-3}$). However, nearshore chlorophyll concentrations and values of PP (see Auger et al., 2015) are lower than satellite-based estimates (Carr, 2001; Gregg et al., 2003). Thus, the cross-shore gradient is not as sharp in the model as in satellite observations. However, SeaWiFS may actually overestimate in situ data in the Mauritanian upwelling. Gregg and Casey (2004) attributed this over-estimation to unmasked Saharan dust in the atmospheric correction algorithm (Moulin et al., 2001). Increased overestimation with increasing chlorophyll concentrations
were also evidenced (Gregg and Casey, 2004). Moreover, model results fall in the range of in situ measurements in the Mauritanian upwelling (Atlantic Meridional Transect, AMT; Aiken et al., 2009; Gibb et al., 2000; Marañon et al., 2000; Pradhan et al., 2006) which reveal lower chlorophyll concentrations than SeaWiFS data. Large phytoplankton cells (diatoms) are generally dominant in the coastal upwelling region (not shown) in agreement with AMT measurements (Aiken et al., 2009). It shows an increasing contribution of smaller cells toward the open ocean. Our model is also able to reproduce this
observed shift in phytoplankton community structure from nearshore to offshore (Gutiérrez-Rodríguez et al., 2011).

The seasonal variability, both in the model and in the data (Fig. 1), is maximum in the Senegalo-Mauritanian region, and minimum off Cape Blanc. Over the Saharan Bank, peaks of plankton productivity occur in spring/summer whereas a relaxation of Trade winds globally induces a lower production in fall. In contrast, plankton productivity peaks in winter/spring in the Senegalo-Mauritanian region. As described from satellite observations (Lathuilière et al., 2008), the
surface chlorophyll maximum is confined to the coast north of Cape Boujdour. The offshore extension of chlorophyll then increases equatorward from the Saharan Bank to Cape Blanc in summer, and to the Senegalo-Mauritanian region in winter. Noticeably, maximum offshore extension is found year round off Cape Blanc (21° N). South of Cape Blanc, maximum offshore extension occurs when nearshore chlorophyll concentrations are maximum in winter, whereas the contrary is found north of Cape Blanc.



### 2.3 Box analysis

In order to explicit the factors controlling the meridional variability of primary productivity off the NW African coast, we carried out a box analysis focusing on nitrate (the main limiting nutrient) and phytoplankton carbon budgets (12–27° N, see Fig. 1) based on a climatology of model outputs over our simulation period (1980–2009). First, we distinguished between the

5 coastal region (from the coast to 0.5° offshore) and the offshore region (covering 4° of longitude further offshore). Second, based on near-surface horizontal circulation patterns in the coastal region, offshore export and bathymetric considerations, the study domain was split into five latitudinal bands. The vertical extension of the ten boxes was chosen from the free surface down to 100m (or the bottom in areas shallower than 100m) to encompass the euphotic layer.

Starting from the northern part of our simulated domain, the circulation off the **Saharan Bank (21–27° N)** is generally

characterized by year round equatorward velocities. Moreover, strong offshore velocities differentiate the **northern Saharan Bank (NSB, 24–27° N)** from the **southern Saharan Bank (SSB, 21–24° N)**.

On the opposite, in the southernmost part of our domain, the **Senegalo-Mauritanian region (12–19° N)** is characterized by a seasonal reversal of meridional velocities with southward/northward direction in winter/summer, respectively, and by enhanced westward velocities in winter. Noticeably, we separated the **southern Senegal region (SS, 12–15° N)** from the rest

of the **Senegalo-Mauritanian region (SM, 15–19° N)** because the circulation patterns differ significantly and the continental shelf is wider south of Cape Verde implying different coastal dynamics.

At the frontier of the two previous main zones, the **Cape Blanc area (CB, 19–21° N)** including the Arguin Bank is globally the place of a meridional convergence of surface water masses and strong offshore velocities.

In these boxes, meridional wind speed and wind curl were averaged to respectively compare upwelling-favourable forcings

and Ekman pumping between boxes. In like manner, the horizontal and vertical velocities and advective/diffusive fluxes of nitrate and phytoplankton biomass (mol m$^{-2}$ s$^{-1}$) were averaged over the faces of each box and the phytoplankton biomass and PP (new and regenerated) were averaged and compared between boxes. Net biological rates (biological source minus sink) in each box were also computed to offer an integrated view of the source and sink terms of phytoplankton biomass and nitrate in each box (mol s$^{-1}$)(i.e. the respective contribution of the advective/diffusive fluxes at the box boundaries and the net

biological rate). Additionally, the average residence time of upwelled water masses in each box was derived from a lagrangian tracking of passive particles based on 3D current fields from the physical model (see full description in Auger et al., 2015). The meridional variability of the ecosystem functioning then could be fully characterized.





# 3 Results

## 3.1 Meridional variability of wind forcings, ocean response and primary productivity in the coastal region

### 3.1.1 Wind forcing, current velocity and nitrate fluxes

Mean annual coastal wind forcing imposed in our simulation, i.e. meridional wind intensity and wind curl, are presented in

Fig. 2 (a & b). Equatorward upwelling-favourable (Fig. 2a) wind is maximum in the southern Saharan Bank and off Cape Blanc. Wind curl shows a clear maximum off Cape Blanc but a weak meridional variability (Fig. 2b).

Mean annual current velocities (vertical and horizontal) on the edge of coastal boxes are presented in Fig. 2 (c & d-e-f). Coastal upwelling shows a strong latitudinal variability with a clear weakening southward (Fig. 2c). Maximum upwelling intensity is by far found over the northern Saharan Bank which is under the influence of particularly active upwelling cells

found in Cape Boujdour and Cape Juby areas (Arístegui et al., 2004; Barton et al., 2004). Minimum upwelling intensity is found off Cape Blanc.

Cross-shore velocities are everywhere directed offshore with maxima off the northern Saharan Bank and Cape Blanc (Fig. 2f) where advection by filaments is the most active (Barton et al., 2004; García-Muñoz et al., 2005; Karakaş et al., 2006).

Offshore transport of nitrate is found maximum in the Cape Blanc and Senegalo-Mauritanian regions. Noticeably, it is significantly higher off Cape Blanc than in the northern Saharan Bank despite equivalent cross-shore velocities.

Southward circulation is found over the Saharan Bank as the signature of a strong upwelling-induced coastal jet (Fig. 2d-e). On the contrary, the northward circulation found in the Senegalo-Mauritanian region represents the eastern branch of the large scale cyclonic circulation characterizing the region of the recirculation gyre (Mittelstaedt, 1991). The Cape Blanc area

is thus characterized by the average meridional convergence of water masses from the Saharan Bank and Mauritania, and this occurs together with minimum coastal upwelling intensity.

Mean annual nitrate transport (vertical and horizontal) through the side of coastal boxes are presented in Fig. 3 (a & b-c-d). Vertical (upwelling-induced) nitrate supply is maximum in the northern Saharan Bank and the Senegalo-Mauritanian region (Fig. 3a), and inversely minimum off Cape Blanc according to minimum upwelling intensity (see above). However, the

southward weakening of upwelling intensity is not reflected in upwelling-induced nitrate supply. Indeed, the vertical nitrate supply is higher south of Cape Blanc than in the southern Saharan Bank contrasting with upwelling-favourable wind intensity.

### 3.1.2 Primary production, phytoplankton biomass and phytoplankton fluxes

The meridional coastal distribution of the annual Primary Production (PP) shows a well marked maximum off Cape Blanc

(Fig. 4a). Similar levels of PP in the boxes, respectively, north and south of Cape blanc are simulated.

The annual mean new production follows the upwelling-induced nitrate supply (Fig. 3a and 4a), except off Cape Blanc where maximum new production is associated to minimum vertical nitrate supply. On the contrary, regenerated production



(ammonium consumption, Fig. 4a) shows a meridional structure opposed to the one seen in upwelling-induced nitrate supplies (Fig. 3a) with higher levels in the southern Saharan Bank, off Cape Blanc (although showing the minimum f-ratio; see Fig. 4b), as well as in the southern Senegal coastal box (Fig. 4a). This agrees with the meridional variability of the ammonium production by microbial remineralization (Fig. 5a) which brings 60–70 % of the ammonium (not shown).

Microbial remineralization depends on detritus concentration, and then potentially on lateral inputs of organic matter. Zooplankton excretion is also minimum in the northern Saharan Bank with a significant overall contribution of mesozooplankton (20–30 %, not shown) and to a lesser extent microzooplankton (5–15 %, not shown). Noticeably, in all coastal boxes except the northern Saharan Bank, PP is mostly related to regenerated production (Fig. 4b; f-ratio<0.5).

Phytoplankton biomass, averaged over the 100 meters depth of the boxes, mirrors that of PP with a maximum off Cape Blanc (Fig. 4c). However, phytoplankton exhibits higher biomass north of Cape Blanc than south which does not translate in higher PP in the northern boxes. (i.e. North Saharan Bank & South Saharan Bank). Phytoplankton biomass in the Senegalo-Mauritanian region and the North Saharan Bank are similar while PP is different. Noteworthy, the phytoplankton biomass is found maximum off Cape Blanc and the South Saharan Bank contrasting with minimum upwelling-induced nitrate supplies

(Fig. 3a).

Off Cape Blanc, phytoplankton biomass is mostly exported through the 100m bottom of the box (Fig. 3e). Otherwise, the meridional distribution of vertical phytoplankton export is weak. The meridional variability of horizontal phytoplankton fluxes (Fig. 3f-g-h) is nearly the same as current velocities (Fig. 2d-e-f), which contrasts with nitrate. Maximum transport of phytoplankton is directed southward and occurs in coastal boxes north of Cape Blanc. In particular, the southern Saharan

Bank and Cape Blanc boxes are characterized by strong horizontal inputs of phytoplankton through their northern boundaries. In the Senegalo-Mauritanian region, the southward fluxes of phytoplankton biomass are interestingly opposed to the northward flux of nitrate. Zonal transport of phytoplankton is directed offshore with a maximum off Cape Blanc and in the northern Saharan Bank and shows maxima off Cape Blanc and in the northern Saharan Bank (Fig. 3h). This latter feature contrasts with nitrate fluxes (Fig. 3d).

**3.1.3 Meridional distribution of the processes controlling the coastal nitrate concentration and phytoplankton biomass**

The annual mean contribution of each source and sink terms (i.e. advective/diffusive tracer fluxes and net biological rate) to the total rate of change of, respectively, nitrate concentration and phytoplankton biomass in coastal boxes are presented in Fig. 6 (a & b). From northern to southern boxes, the functioning in each box is described. Note that the contribution of

diffusion, i.e. horizontal and vertical mixing, is negligible compared to advection.

In the northern Saharan Bank, the nitrate supply seems to be dominated by coastal upwelling (+36 %) ,which exceeds nitrate advection from the north (+13 %). The major sink is found to be horizontal advection (-29 %), i.e. southward (-16 %) and westward advection (-13 %, offshore advection), actually exceeding the biological uptake (-21 %). The net phytoplankton





local growth (+33 %) exceeds the transport of phytoplankton by advection (+17 %) through the northern boundary of the northern Saharan Bank box. Phytoplankton is then exported offshore (-25 %) and southward (-21 %) and to a much lesser extent through sedimentation below 100m (-4 %).

In the southern Saharan Bank, the nitrate supply is shared between advection through the northern boundary from the northern Saharan Bank (+26 %) and coastal upwelling (+23 %). Southward advection to the Cape Blanc area (-29 %) is then the main nitrate sinks with both contributions from the biological sink (-15 %) and offshore export (-6 %) being reduced compared to their role in the northern Saharan Bank. In the southern Saharan Bank, the net local phytoplankton growth is not anymore the main source of phytoplankton (+17 %) when compared to the transport of phytoplankton from the northern Saharan Bank (+32 %). Phytoplankton biomass is then exported southward (-26 %) and offshore (-19 %), the vertical export below 100m depth still being low (-5 %).

Off Cape Blanc, as in the southern Saharan Bank, the nitrate budget is mostly driven by horizontal advective fluxes. The nitrate supply is mostly due to transport through the northern boundary (+18 %) and the southern boundary (+25 %) from, respectively, the northern Saharan Bank and the Senegal-Mauritanian region. Most of the nitrate is then exported offshore (-41 %). The respective contributions of the coastal upwelling source (+6 %) and the biological sink (-9 %) are of minor importance. As in the southern Saharan Bank, phytoplankton biomass is mostly enhanced by southward advection from the southern Saharan Bank (+28 %) exceeding net biological production (+22 %). Noticeably, this region displays the higher offshore export of phytoplankton (-39 %). Contributions of southward and vertical transport are weak (respectively -5 %. and -6 %).

In the Senegalo-Mauritanian region, the nitrate supply is shared between coastal upwelling (+27 %) and advection from southern Senegal (+23 %). Nitrate sinks are almost equally distributed between offshore export (-18 %), northward advection to Cape Blanc (-17 %) and biological activity (-15 %). In this region, phytoplankton increases mostly because of net local growth (+44 %). Phytoplankton transport from the north (+6 % ) and toward the south (-3 %) are of minor importance. Phytoplankton is primarily exported offshore (-41 %).

Finally, in southern Senegal, nitrate is mostly supplied by advection (+33 %) from the southernmost boundary of our boxes and disappears due to northward advection (-32 %) toward the Senegalo-Mauritanian region. Contributions of the coastal upwelling source (+16 %) and the biological sink (-12 %) are of the same magnitude while the offshore export (5 %) is much less important. Phytoplankton fluxes are very similar to those found in the SM region with a main source related to net local growth (+41 %) and a main sink through offshore export (-43 %). Phytoplankton transport from the north (+4 % ) and toward the south (-5 %) are of minor importance.

A tentative schematic representing the main fluxes for nitrate and phytoplankton is given in Fig. 7. It shows that coastal upwelling of nitrate, despite being significant in all regions but the Cape Blanc area, is the dominant supplier only in the northern Saharan Bank and the Senegalo-Mauritanian region. In all other regions, nitrate supply is dominated by meridional advection. Indeed, the southern Saharan Bank is mostly fuelled by nitrate advected southward from the northern Saharan



Bank while the Cape Blanc area gets nitrate inputs from the southern Saharan Bank and the Senegalo-Mauritanian region. South of this frontal zone in the 2 southernmost boxes (i.e. SM and SS), northward nitrate transport becomes a key player in the nitrate budget. Net local phytoplankton growth is the most important source of phytoplankton in three of our 5 boxes (i.e. the NSB, the SM and the SS) but this prevalence is mostly marked only in our 2 southernmost boxes (the SM and SS boxes). The three north boxes display close contributions of the phytoplankton supply from northward advection and the net local growth with the latter only dominating in the northernmost box (NSB).

## 3.2 Meridional variability of spring wind forcing, ocean response and primary productivity in the offshore region

### 3.2.1 Wind forcing, current velocity and nitrate fluxes

The offshore extension of chlorophyll has been shown to display a marked seasonal variability with a maximum in spring (Lathuilière et al., 2008; see Fig.1). In the offshore region, the chlorophyll variability depends on the export of coastal productivity. Additionally, the wind stress can be responsible for vertical mixing that enhances the exchanges of inorganic and organic matter between the euphotic and aphotic layers. The vertical nutrient supply to the enlightened surface layer and the phytoplankton export below the euphotic layer may also be enhanced by positive/negative Ekman pumping, respectively linked to positive/negative wind stress curl. In order to explicit the offshore extension in spring of the rich phytoplankton pattern, mean wind forcings from March to May (i.e. wind intensity and wind curl) are first presented in Fig. 8 (a & b). During these months, the wind intensity increases from the northern Saharan Bank to Cape Blanc (where it peaks) and then decreases southward (Fig. 8a). Alternatively, wind stress curl shows a monotonous southward increase (Fig. 8b) from the northern Saharan Bank to southern Senegal with negative values from the NSB to the Cape Blanc region.

During spring, the vertical velocities at the bottom of the offshore boxes display the same meridional structure (Fig. 8c) as the the wind curl. As a proxy of offshore export from the coastal band, inward velocities at the eastern boundary are found maximum off the northern Saharan Bank and off Cape Blanc (Fig. 8g), and minimum off the southern Saharan Bank. Noteworthy, at the western boundaries, velocities are directed offshore except off the northern Saharan Bank where shoreward intrusions from the subtropical gyre are detected (Fig. 8f). Maximum offshore velocities happen to be found at the latitude of the Cape Blanc region. Southward velocities are found off the southern Saharan Bank and off Cape Blanc (Fig. 8e) while northward velocities are found off southern Senegal (Fig. 8d). Off the Senegalo-Mauritanian region, inward velocities at both north and southern boundaries indicate the presence of a convergence zone in this latitudinal range during spring (Fig. 8d-e).

Mean spring vertical nitrate fluxes (advection and diffusion) at the bottom of offshore boxes are presented in Fig. 9 (a & b). A striking result is that vertical advection falls in the same order of magnitude than diffusion fluxes with an opposite meridional variability. On the one hand, vertical nitrate supply by advection is found in all offshore boxes (but weak for the NSB) except in the latitudinal band of the SSB region (Fig. 9a). Noteworthy, vertical nitrate supply by advection off the northern Saharan Bank and off Cape Blanc is found while vertical velocities are pointing downward. On the other hand, the





vertical nitrate supply due to turbulent diffusion shows a clear southward decrease (Fig. 9b). It is actually stronger off the Saharan Bank (North and South) and Cape Blanc than off Mauritania and Senegal.

The patterns of lateral nitrate fluxes during spring (Fig. 9c-d-e-f) mainly follow current velocities but with significant deviations (see Fig. 8d-e-f-g). Maximum alongshore fluxes are found south of Cape Blanc and are directed northward (Fig.

9c-d). Nitrate inputs from the coastal band (eastern boundary) increase southward compared to cross-shore currents, especially off the Senegalo-Mauritanian region (Fig. 9f), which is attributed to a southward increase of nitrate concentrations in upwelling source waters. The maximum off Cape Blanc is smoothed compared to current velocities (Fig. 8g) indicating a relative nutrient depletion of coastal waters off Cape Blanc.

### 3.2.2 Primary production, phytoplankton biomass and phytoplankton fluxes in spring

Annual mean PP (new and regenerated production), f-ratio and phytoplankton biomass are presented in Fig. 10. According to the meridional pattern of offshore extension of chlorophyll in spring (Lathuilière et al., 2008), offshore PP and phytoplankton biomass are found maximum off Cape Blanc and significantly higher in the Senegalo-Mauritanian region than off the Saharan Bank. Both new and regenerated production display the same meridional variability. Nevertheless, regenerated production is generally more intense (except off the northern Saharan Bank) and also more variable in space compared to

new production. This corresponds to the meridional variability of ammonium production by both microbial remineralization and zooplankton excretion (see Fig. 5b) which respectively contribute to 75 % and 30–40 % (15–20 % for both micro- and mesozooplankton, not shown). The meridional variability of PP is then controlled first by organic matter inputs from the coastal band which stimulate regenerated production and, in less manner, by local zooplankton excretion; second by nutrient inputs from the coastal band responsible for new production.

Mean spring vertical phytoplankton fluxes (advection and diffusion) at the bottom of offshore boxes are presented in Fig. 11 (a & b). A striking result is that diffusion fluxes exceed advection fluxes by one order of magnitude. Turbulent diffusion is overall responsible for vertical export off the Saharan Bank, and to a lesser extent off Cape Blanc (Fig. 11b). Moreover, the offshore vertical export of phytoplankton biomass due to advection exclusively occurs off the Saharan Bank and off Cape Blanc (Fig. 11a). The meridional variability of vertical advection is indeed driven by wind curl (see Fig. 8). As a result, the

total offshore vertical export is clearly maximum off the Saharan Bank which is also a sign of maximum dilution of phytoplankton biomass over the water column in this region.

Mean spring phytoplankton horizontal advective fluxes at the boundaries of offshore boxes are finally presented in Fig. 11c-d-e-f. The meridional variability of lateral phytoplankton fluxes is nearly the same as current velocities (see Fig. 8d-e-f-g). Nevertheless, the westward fluxes through the eastern boundary are more important than expected from lateral velocities

from Cape Blanc to Senegal.



### 3.2.3 Which processes control the offshore nitrate concentration and phytoplankton biomass ?

Annual mean contribution of each source and sink terms of nitrate concentration and phytoplankton biomass in offshore boxes, i.e. advective/diffusive tracer fluxes and net biological rate, are presented in Fig. 12 (a & b). From northern to southern boxes, the functioning in each box is described. Note that the contribution of horizontal diffusion is negligible compared to vertical diffusion and advection.

In the northern Saharan Bank, the nitrate supply is equally due to coastal inputs and vertical mixing (+22 % and +20 %, resp.), the contribution of nitrate advection from the north (+1 %) and offshore upwelling (+3 %) remaining insignificant. At the same time, the major sink is found to be the biological activity (-43 %), exceeding southward advection (-11 %). Alternatively, in the southern Saharan Bank, the nitrate supply is shared between vertical mixing (+18 %) and northerly advection from the northern Saharan Bank (+13 %) exceeding southerly advection (+7 %). Nitrate then mostly disappears due to biological activity (-52 %). Off Cape Blanc, the nitrate supply is mostly due to coastal inputs (+34 %), and then removed by biological activity (-35 %) and offshore export (-13 %). In the Senegalo-Mauritanian region, the nitrate supply is shared between coastal inputs (+28 %), northerly advection from southern Senegal (+16 %) and bottom advection (+9 %). In like manner, the nitrate sink is more or less equally distributed between biological activity (-28 %) and westward advection (-16 %). Finally, in southern Senegal, nitrate is mostly supplied by southerward advection (+43 %) and disappears due to lateral advection (-32 %), distributed between westward and northward advection (-19 % and -13 %, resp.), exceeding the biological activity (-14 %).

The source of phytoplankton biomass in the northern Saharan Bank is equally distributed between net biological activity (+23 %) and coastal inputs (+23 %). Phytoplankton biomass is then mostly removed by vertical export through vertical mixing (-36 %) and southward transport (-12 %). Alternatively, in the southern Saharan Bank, phytoplankton biomass is mostly due to lateral advection (+31 %), originating from the north (+18 %) and from the coast (+13 %), followed by biological activity (+14 %). Phytoplankton biomass is still mostly exported vertically through vertical mixing (-36 %). Off Cape Blanc, the phytoplankton biomass mainly results from coastal inputs (+43 %) and removed through lateral advection (-30 %), i.e. offshore (-20 %) and southward (-10 %), which exceeds vertical mixing (-9 %) and biological activity (-8 %). Finally, in the Senegalo-Mauritanian and southern Senegal regions, phytoplankton biomass is mostly enhanced by coastal inputs (+37 % and +32 %, resp.) and disappears through biological activity (-33 % and -29 %, resp.). Southward inputs have also a noticeable contribution off southern Senegal (+13 %).

To summarize (see Fig. 13), off the Saharan Bank, nitrate is equally supplied by vertical mixing and lateral advection whether from the coast in the northern Saharan Bank or from the north in the southern Saharan Bank. As a consequence, phytoplankton biomass results from net biological production and lateral advection. Phytoplankton biomass almost exclusively disappears through vertical mixing. Alternatively, south of the Saharan Bank, the nitrate supply is dominated by lateral advection whether from the coast off Cape Blanc, from the coast and the south off the Senegalo-Mauritanian region, and from the south off southern Senegal. In these regions, the phytoplankton biomass is mostly enhanced by zonal advection





and disappears through a negative net biological rate. Indeed, the corresponding boxes were defined in the transition zone between eutrophic coastal waters and oligotrophic waters of the subtropical gyre where phytoplankton communities collapse through mortality and grazing. Off Cape Blanc, zonal advection however dominates due to stronger nutrient and phytoplankton inputs. Thus, the collapse of phytoplankton communities is expected further offshore.

## 4 Discussion

In this study, we investigated the processes driving the meridional variability of phytoplankton biomass and PP in coastal and offshore regions. We will first discuss the sensitivity of coastal upwelling to the wind forcing, which is a key player for the vertical nutrient supply in the coastal region. Then we will focus on the meridional variability of coastal phytoplankton biomass and PP (new and regenerated production) in relation with the transport of matter along the coastal band and to the open ocean. Finally, we will seek to explicit the processes driving the meridional variability of the offshore extension of the coastal chlorophyll pattern during the spring upwelling season.

### 4.1 Sensitivity of coastal upwelling to the wind forcing

In our simulation, the meridional variability of coastal upwelling is not correlated to the local variability of wind-driven Ekman transport and Ekman pumping. This result questions the estimation of vertical velocities based on local wind forcing that were commonly used in EBUS (Bakun, 1990; Lathuilière et al., 2008; Messié et al., 2009; Messié and Chavez, 2014). It appears that coastal upwelling depends on many other factors including the large scale dynamical state of the ocean and the coastal geomorphology (Benazzouz et al., 2014; Mason et al., 2012). As an example, the upwelling limitation by onshore geostrophic flow have been shown to play a key role in driving some coastal upwellings (Marchesiello and Estrade, 2010; Messié and Chavez, 2014). Local and large scale processes, while not entirely decoupled (e.g. NECC intensity and the seasonal weakening of trade winds are coupled; Mittelstaedt, 1991), act at different time scale and impact in different ways the coastal upwelling. The simulated spatial and temporal variability of surface circulation are in good agreement with the satellite-tracked drifters (see Section 2.2). Therefore it gives us confidence that the model can be used to explicit the factors responsible for the sensitivity of coastal upwelling to the wind forcing. For this latter purpose, we further analyze the seasonal cycles of meridional wind versus vertical and horizontal advection at the edge of coastal boxes (Fig. 14).

In the northern part of our domain, the presence of the Cape Boujdour Filament (Barton et al., 2004; García-Muñoz et al., 2004; Karakaş et al., 2006) lead to a year round strong offshore export (associated with a strong cross-shore divergence) in the northern Saharan Bank (Fig. 14c). In the southern Saharan Bank and off Cape Blanc, upwelling-favourable wind increases during spring and/or early fall (Fig. 14a). This induces an acceleration of the equatorward jet from the northern Saharan Bank to the Cape Blanc area (Fig. 14d) which tends to create a meridional divergence promoting coastal upwelling over the northern Saharan Bank.





Southward, during the spring/summer and fall/winter transitions, the poleward geostrophic circulation which establishes from southern Senegal to Cape Blanc (Fig. 14d; Lázaro et al., 2005; Mittelstaedt, 1991; Stramma et al., 2005; Wooster et al., 1976) limits the southward extension of the equatorward jet found over the Saharan Bank. It creates a meridional convergence of coastal water masses Saharan Bank in the Cape Blanc area (Fig. 14d) which limits significantly the intensity

of the coastal upwelling (Fig. 2c). Indeed, a downwelling period is paradoxically found in summer off Cape Blanc when upwelling-favourable wind is maximum (Fig. 14a-b). Downwelling occurs as the cold and dense upwelling water from the Saharan Bank encounters the warm and stratified equatorial water from the NECC (Mittelstaedt, 1991). Noticeably, the response of the Cape Blanc filament (spring and fall peaks of cross-shore velocities) is delayed by one month compared to the alongshore jet peaks. This suggests that the local downwelling (driven by meridional convergence) and the inertia of the

upwelling jet over the Saharan Bank (as described by Benazzouz et al., 2014) are the primary drivers of the response of the Cape Blanc filament to the wind forcing. In the southernmost part of our domain, the Senegalo-Mauritanian region, the seasonal cycle of the coastal upwelling is likewise partly driven by the equatorward wind intensity and a downwelling which is detected in late spring while the equatorward wind is weakening (but is not yet minimum).

In the NW African region, coastal topography effects and alongshore geostrophic flow (related to large scale circulation

patterns) noticeably influence the convergence/divergence of coastal water masses. They modulate the coastal divergence driven by the Ekman transport, i.e. the response of coastal upwelling to the wind forcing.

## 4.2 Meridional variability of coastal phytoplankton biomass, primary production and matter transfers

Albeit PP is mainly regulated by new production and the amount of nitrate supply by wind-driven upwelling (Ohde and Siegel, 2010), the meridional variability of PP and phytoplankton biomass is noticeably influenced by regenerated

production (fuelled by the uptake of ammonium). High regenerated production is a sign that upwelled nitrate is efficiently used (Lachkar and Gruber, 2011; Messié and Chavez, 2014) as a result of high residence time in the NW African upwelling system (Lachkar and Gruber, 2011). We show that the meridional variability of regenerated production actually deviates from the variability of new production as a result of (1) the lateral advection of ammonium, particulate detritus and dissolved organic matter that are remineralized, and (2) retention patterns increasing the residence times of water masses.

In the northern Saharan Bank, new production and phytoplankton biomass remain relatively low. Low nitrate concentration in upwelling source waters (North Atlantic Central Water; Arístegui et al., 2009) and short residence time (due to high horizontal advection by the coastal upwelling jet, see Fig. 15a) limit the phytoplankton growth. Phytoplankton indeed requires time to complete nutrient uptake (Dugdale et al., 1990; Zimmerman et al., 1987) and low residence times also limit regenerated production leading to minimum PP (Checkley and Barth, 2009; Messié et al., 2009). Upwelled water masses are

then exported southward and offshore. Nitrate is mainly exported southward (coastal water masses) while there is relatively more phytoplankton exported offshore. During synoptic events of coastal upwelling, the coastal jet exports nitrate-rich and phytoplankton-poor water masses to the south. Inversely, the relaxation of the coastal jet enhances residence time and promotes the local building of phytoplankton biomass. The combined effect of this local growth and the high filament





activity around Cape Boujdour (Barton et al., 2004; García-Muñoz et al., 2004; Karakaş et al., 2006) results in an offshore transport of phytoplankton-rich and nitrate-depleted water masses.

New production, fuelled by nitrate upwelled in the north Saharan Bank, happens partly further downstream in the southern Saharan Bank where such remote influence accounts for 50 % of new production. This phenomenon has also been observed

downstream of major other upwelling cells as in the Benguela region for example (Hardman-Mountford et al., 2003).

Off Cape Blanc, the meridional convergence of water masses results in subduction events which enhances the vertical extension of the plankton-rich pattern and lead to high levels of phytoplankton biomass when integrated over the 0-100m surface layer. The phytoplankton biomass is maintained by maximum levels of regenerated production, actually exceeding new production by more than two fold. Usually the regenerated production relies on high residence time favourable to

efficient recycling (see Fig. 15a). However, the water masses residence time in the southern Saharan Bank is low and can not explain the high level of regenerated production. In this region the regenerated production is rather due to organic matter supply off Cape Blanc (Fig. 5a). Note that the meridional variability of secondary production (grazing rate) follows that of PP (bottom-up control, not shown) suggesting a bottom-up control of the phytoplankton biomass rather than a top-down control by zooplankton grazers. Zooplankton excretion instead participates to enhance regenerated production in areas where

the lateral input of plankton biomass is elevated, which is especially the case off Cape Blanc.

Inversely, in the Senegalo-Mauritanian region, only moderate regenerated production is found year round although residence time is relatively high compared to the southern Saharan Bank and Cape Blanc areas. This supports the idea that, in the Cape Blanc and Senegalo-Mauritanian regions, regenerated production is rather driven by the amounts of organic matter supplies through lateral boundaries than by local input due to high residence time. In the Senegalo-Mauritanian region, new

production is enhanced by vertical nitrate supply during the winter/spring upwelling period (South Atlantic Central Water; Arístegui et al., 2009), and by southerly inputs of equatorial nutrient-rich water masses from the Gulf of Guinea in late spring and late fall (Lázaro et al., 2005; Mittelstaedt, 1991). Some authors have also reported the potential impact on primary productivity of horizontal advection of warm, nutrient- and chlorophyll-poor waters by the NECC (Lathuilière et al., 2008) when the trade winds weaken in summer and early fall (Mittelstaedt, 1991; Stramma et al., 2005). Indeed, the weakening of

the trade winds and the advection by the NECC are actually coupled but the transition period might enable transient intrusions of nutrient-rich coastal waters from the Gulf of Guinea. Let us also mention that nutrient loads by rivers, which are not accounted for in our model configuration, may significantly sustain marine productivity during the monsoon summer period as high coastal concentrations of nutrients have been recently observed at the end of the rainy season in southern Senegal (E. Machu, pers. comm.).

At the coast, the meridional variability of new production follows the pattern of vertical nitrate supply, except off Cape Blanc where maximum new production and phytoplanton biomass is mostly related to lateral nitrate injection. This points a gap of the first estimates of nitrate supply by coastal upwelling based on wind-derived vertical velocity and nitrate concentration in upwelling source waters (Gruber et al., 2011; Messié et al., 2009; Messié and Chavez, 2014) which were used to explore the link between nutrient supply and primary productivity in EBUS. Off NW Africa, this method may actually provide



misleading vertical nitrate supply in the northern Saharan Bank (underestimation) and off Cape Blanc (overestimation) (see Section 4.1). In particular, horizontal convergence and subduction of nutrients in late spring/early summer (see above) seem to limit the annual vertical nitrate supply by the upwelling off Cape Blanc. Interestingly, satellite-derived diagnostics of low residence time off Cape Blanc (Messié and Chavez, 2014), confirmed by our model results (see Fig. 15), also suggest that

subduction may play a key role in regulating PP off Cape Blanc. Based on the results of Messié and Chavez (2014), the potential overestimation of vertical nitrate supply identified in this study may also concern other EBUS.

### 4.3 Extension of the coastal-rich phytoplankton pattern in spring/summer

The offshore extension of the coastal surface chlorophyll pattern is highly variable in space and time off NW Africa. As described from SeaWiFS data (Lathuilière et al., 2008), the chlorophyll extension is narrow over the Saharan Bank (less than

10 100 km), wide off Cape Blanc (approximately 200 km) and can reach 400 km at the end of the spring upwelling season in the Senegalo-Mauritanian region. Focusing on the meridional variations, Lathuilière et al. (2008) investigated the potential impact of several physical and biological processes on this offshore extension: (i) the distance of the upwelling front from the coast, the wind stress curl, (ii) the impact of mesoscale and submesoscale dynamics on the cross shelf transport, and (iii) the limitation of phytoplankton growth by nutrients. Our modelling approach was designed to test some of these hypothesis

and to better explicit the mechanisms driving the offshore extension of surface chlorophyll off NW Africa. For this purpose, we focus on the offshore region in the spring period: at this season, maximum coastal upwelling is found off the Saharan Bank (Fig. 2c), but the phytoplankton biomass extension is maximum off Cape Blanc and in the Senegalo-Mauritanian region as attested by the meridional variation of phytoplankton biomass in the offshore region (Fig. 10).

Upward Ekman pumping due to positive wind curl has been reported to contribute significantly to the vertical mass flux

associated to coastal upwelling in a band extending 200 km from the coast (Castelao and Barth, 2006; Enriquez and Friehe, 1995). Off NW Africa, Ekman pumping has been suggested to contribute to half of the surface chlorophyll variability on interannual timescale (Pradhan et al., 2006). Our results indicate downward and upward wind-induced Ekman pumping of respectively north and south of Cape Blanc (Figs. 8b & 9a). However, the contribution of vertical advection to the nitrate input in the surface layer is negligible compared to lateral advection and vertical mixing (Fig. 12a). Thus, the offshore

extension of the phytoplankton biomass is not primarily driven by the nutrient supply due to Ekman pumping, as already suggested over seasonal time scale by Lathuilière et al. (2008).

The vertical mixing contributes significantly to nitrate supply off the Saharan Bank (especially in the southern Saharan Bank, see Fig. 12a). Indeed, in spring, the mixed layer depth can reach more than 100 m off the Saharan Bank as a result of winter convection, while remaining generally less than 60 m in the Senegalo-Mauritanian region (not shown). This is consistent

with the global climatology of the mixed layer depth from de Boyer Montégut et al. (2004). However, such nitrate supply does not translate into high phytoplankton biomass. In fact, the vertical mixing is also responsible for a significant vertical export of phytoplankton biomass below 100 m which may limit the phytoplankton biomass off the Saharan Bank (Huntsman and Barber, 1977). The redistribution of phytoplankton biomass within the mixed layer most likely decreases the surface



phytoplankton biomass detected by satellite north of Cape Blanc (dilution effect) which may participate to explain the weak offshore extension of surface chlorophyll (Lathuilière et al., 2008).

Overall, our study shows that the lateral advection of nutrients and phytoplankton biomass is mostly directed alongshore (southward) off the Saharan Bank and cross-shore south of Cape Blanc. Thus, inputs of nutrients and phytoplankton biomass

from the coast are mainly found south of Cape Blanc (Fig. 9f & 11f). This is consistent with the high regenerated production (Fig. 10) found together with low residence times south of Cape Blanc (see Fig. 15b). Nutrient limitation might then play a minor role in the weak offshore extension of chlorophyll north of Cape Blanc questioning the hypothesis of 2ZOTLathuiliere2008. The advection by filaments is found to be a major process enhancing cross shelf transport off Cape Blanc (Kostianoy and Zatsepin, 1996). Likewise, high kinetic energy in the Senegalo-Mauritanian region suggests

thatfilaments and mesoscale eddies may enhance cross shelf transport (see Lathuilière et al., 2008). In this region, the coastal upwelling jet develops inshore of a large scale northward circulation, and the resulting current shear may help the development of baroclinic instabilities mostly responsible for eddy generation in upwelling systems (Capet et al., 2008). In contrast, off the Saharan Bank, the coastal upwelling jet flows southward together with the Canary Current which may limit the development of such mesoscale activity. The greater offshore extension of the coastal phytoplankton biomass in the

Senegalo-Mauritanian region may then be primarily explained by the lateral advection of nutrients and phytoplankton biomass, as already suggested by Lathuilière et al. (2008). Nevertheless, our model results indicate that the nutrient input is not only from the coastal region as we identified a significant impact of transient southern intrusions of nutrient-rich waters in the Senegalo-Mauritanian region originating from the Guinean upwelling (Lázaro et al., 2005; Mittelstaedt, 1991).

## 5 Conclusion

In the present study, a physical-biogeochemical modelling approach seeks to provide a first mechanistic understanding of the drivers of the seasonal variability of the primary productivity spatial distribution in the NW African upwelling system. To this aim, a comparative box analysis representing homogeneous sub-regions in terms of near-surface horizontal circulation was conducted. Our physical-biogeochemical simulation reproduced accurately observed patterns of surface ocean circulation and chlorophyll in our region of interest. We then analysed the distribution (and its variability) of phytoplankton

biomass and production in regards to advective and diffusive fluxes of nutrients and phytoplankton at the box boundaries.
Our results suggest that coastal topography effects and alongshore geostrophic flow related to large scale circulation patterns modulate the coastal divergence driven by the Ekman transport. The effect of wind is amplified off Cape Boujdour and dampened off Cape Blanc. Coastal upwelling of nitrate, despite being significant in all regions but the Cape Blanc area, is the dominant supplier only in the northern Saharan Bank and the Senegalo-Mauritanian region. Elsewhere, nitrate supply is

30 dominated by meridional advection. Thus, the meridional variability of new production follows that of vertical nitrate supply, excepted off Cape Blanc where maximum new production is mostly related to lateral nitrate injection from the northern or southern part of our domain depending on seasonality. Net local phytoplankton growth is the exclusive driver of



phytoplankton biomass only in the Senegalo-Mauritanian region. North of Cape Blanc, the phytoplankton supply from northward advection becomes as important as the net local phytoplankton growth with the latter only dominating off Cape Boujdour. The phytoplankton biomass is also maintained by high levels of regenerated production exceeding new production by more than two fold off Cape Blanc in particular. While the regenerated production relies on high residence time favourable to efficient recycling in the southern Saharan Bank, regenerated production is more impacted by the amounts of organic matter supplies through lateral boundaries than by residence time in the Cape Blanc and Senegalo-Mauritanian regions.

As previously suggested by Lathuilière et al. (2008), the offshore extension of the phytoplankton biomass in spring, more pronounced south of Cape Blanc, is not driven by the nutrient supply due to Ekman pumping. We additionally show that, off the Saharan Bank, the vertical mixing is responsible for a significant vertical export of phytoplankton biomass below 100 m which is not the case south of Cape Blanc. Besides, the redistribution of phytoplankton biomass within the mixed layer may artificially decrease the surface phytoplankton biomass detected by satellite north of Cape Blanc (dilution effect). Overall, the lateral advection of nutrients and phytoplankton biomass is mostly directed alongshore (southward) off the Saharan Bank and cross-shore south off Cape Blanc. Nutrient limitation due to low nutrient concentrations in upwelling source waters might then only play a minor role in the weak offshore extension of surface chlorophyll north of Cape Blanc. The greater offshore extension of phytoplankton biomass in the Senegalo-Mauritanian region then effectively results from a lateral advection of coastal nutrients and phytoplankton biomass. Nevertheless, the nutrient input is not only from the coast as transient southern intrusions of nutrient-rich waters from the Guinean upwelling may significantly fertilize the Senegalo-Mauritanian region.

Future studies should investigate the response in primary productivity to the intra-seasonal and event-scale variability of wind-induced coastal upwelling and nutrient inputs at the box boundaries, and its impact on interannual variability. The year-to-year evolution of fish stocks and migrations may greatly depend on changes in the physical and biogeochemical conditions. This could be tested using our modelling approach by comparing our model inter-annual variability with estimations of fish abundance in the NW African region. Understanding the processes which drive seasonal and inter-annual variability of the upwelling region also represents a first step towards a robust projection of the effect of climate change on the biogeochemistry of the region and therefore on the halieutic resources.

### Acknowledgements

We are deeply indebted to Pierrick Penven, Xavier Capet and Elodie Gutknecht for methodological discussions. We also wish to thank Tristan Le Toullec and computer engineers from the Institut Universitaire Européen de la Mer (IUEM, Brest, France), particularly Emmanuel Taboré, for their technical support. The present work was supported by the Franco-Moroccan EPURE project (Eléments trace métalliques, Pollution, Upwelling et Ressources – http://anr-epure.net) under the call CEP&S of the French National Research Agency (ANR), the AWA project (Ecosystem Approach to the management of



fisheries and the marine environment in West African waters – http://www.awa-project.org), the FP7 PREFACE project (No. 603521, Enhancing prediction of Tropical Atlantic climate and its impacts – http://preface.b.uib.no) and the LabexMER (No. ANR-10-LABX-19-01, http://www.labexmer.eu). Open boundary conditions for our regional simulation were provided through by the FP7 EURO-BASIN project (No. 264933). Numerical simulations were performed using HPC resources from

5    CAPARMOR (CAlcul PARallèle Mutualisé pour l'Océanographie et la Recherche), a cluster hosted at Ifremer (Brest, France). Additional support during the writing phase was provided by the Instituto Milenio de Oceanografia (IMO-Chile), funded by the Iniciativa Cientifica Milenio (ICM-Chile).

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



**Figure 1: Seasonal climatology of sea surface chlorophyll concentrations (background) from SeaWiFS satellite data (1998–2009) and near-surface currents (vectors) from the Global Drifter Program (1979–present, Lumpkin and Johnson, 2013) in (a) winter (january–march) and (b) summer (july–september). Same for ROMS-PISCES in (c) winter and (d) summer. The ten boxes used in this study are superimposed (black boxes). Main surface currents and deep water masses over the study area are presented over a map of simulated surface chlorophyll averaged over the SeaWIFS period (e). NEC: North Equatorial Current; NECC: North Equatorial Countercurrent; CVFZ: Cape Verde Frontal Zone; NACW: North Atlantic Central Water; SACW: South Atlantic Central Water..**




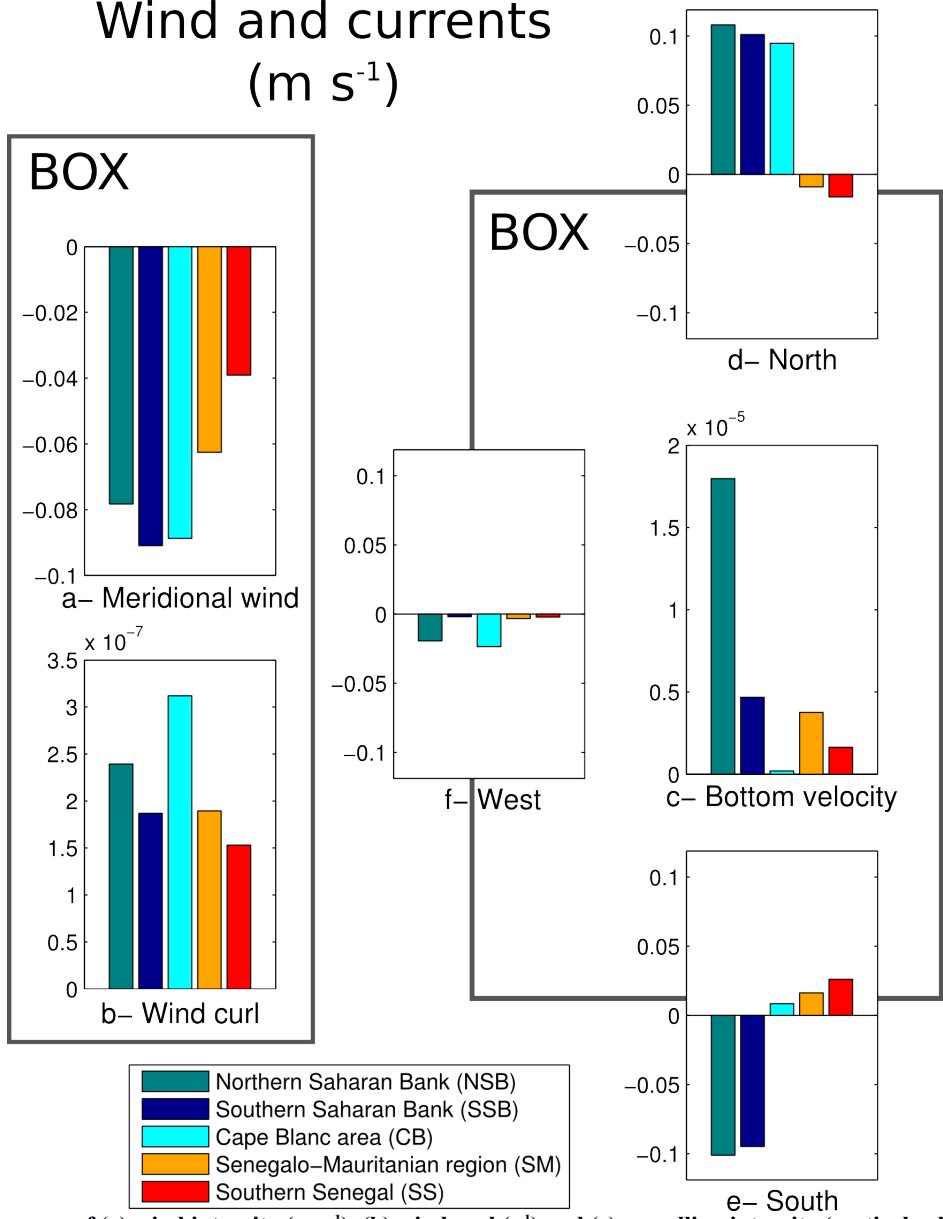

**Figure 2: Annual average of (a) wind intensity (m s⁻¹), (b) wind curl (s⁻¹) and (c) upwelling intensity (vertical velocity at the bottom, in m s⁻¹) within coastal boxes and lateral velocities at the (d) northern, (e) southern and (f) western boundaries (m s⁻¹). Vertical and lateral velocities are defined positive inward (so vertically upward). Each color corresponds to a box (see legend).**





**Figure 3: Annual average of (a) vertical nitrate transport at the bottom of coastal boxes and (b-c-d) lateral transport at the (b) northern, (c) southern and (d) western boundaries (molN m⁻² s⁻¹); annual average of (e) vertical phytoplankton transport at the bottom of coastal boxes and (f-g-h) lateral transport at the (f) northern, (g) southern and (h) western boundaries (molC m⁻² s⁻¹). Vertical and lateral transports are defined positive inward (so vertically upward). Each color corresponds to a box (see legend). Note that the transport by vertical diffusion is one order of magnitude lower compared to advection in coastal boxes.**





**Figure 4: Annual average of (a) primary production (PP, gC m⁻³ year⁻¹), new production (gC m⁻³ year⁻¹) and regenerated production (gC m⁻³ year⁻¹), (b) f-ratio (new/regenerated production) and (c) phytoplankton biomass (mgC m⁻³) in coastal boxes. Each color corresponds to a box (see legend).**





**Figure 5: Remineralization rate of organic carbon (gC m$^{-3}$ year$^{-1}$) through microbial activity and zooplankton excretion (micro-and mesozooplankton) (a) averaged annually in coastal boxes and (b) over the spring period (March–May) in offshore boxes. Each color corresponds to a box (see legend).**





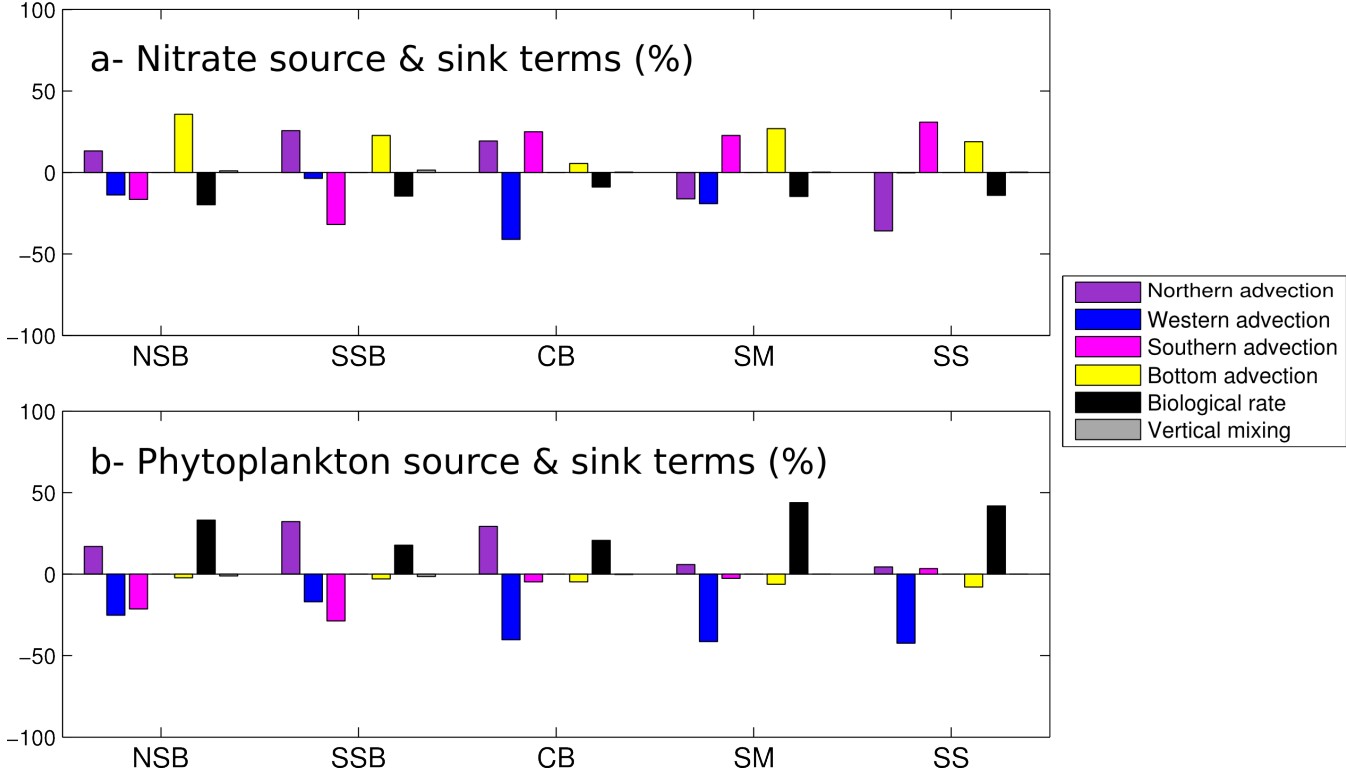

**Figure 6: Annual average contribution of the different source and sink terms of (a) nitrate and (b) phytoplankton biomass in coastal boxes ( %, positive inward): northern, western and southern horizontal advection, vertical advection and diffusion (vertical mixing) at the bottom and net local biological rate of change.**



**Figure 7:** Schematic of the annual average contribution of the different source and sink terms of nitrate and phytoplankton concentration within each box defined in this study. Each color corresponds to a box (see legends in Fig. 2 for coastal and Fig. 8 for offshore boxes). Arrows indicate horizontal advection, circles indicate vertical advection, squares indicate vertical mixing and triangles indicate biological processes. Within a circle or square, a white point indicates a source while a white cross indicates a sink; biological processes are a source/sink if a triangle heads upward/downward, respectively. The size of arrows, circles, squares and triangles indicates the magnitude of the contribution of each source/sink term. For coastal boxes, the information is equivalent to that given in Fig. 6.





**Figure 8:** Spring average (March–May) of (a) wind intensity (m s⁻¹), (b) wind curl (s⁻¹) and (c) upwelling intensity (vertical velocity at the bottom, in m s⁻¹) within offshore boxes and lateral velocities at the (d) northern, (e) southern, (f) western and (g) eastern boundaries (m s⁻¹). Vertical and lateral velocities are defined positive inward (so vertically upward). Each color corresponds to a box (see legend).





**Figure 9: Spring average (March–May) of vertical nitrate transport at the bottom of offshore boxes by (a) advection and (b) diffusion (vertical mixing) and (c-d-e-f) lateral transport at the (c) northern, (d) southern, (e) western and (f) eastern boundaries (defined positive inward, so vertically upward) (molN m$^{-2}$ s$^{-1}$). Each color corresponds to a box (see legend).**





**Figure 10: Spring average (March–May) of (a) primary production (PP, gC m⁻³ year⁻¹), new production (gC m⁻³ year⁻¹) and regenerated production (gC m⁻³ year⁻¹), (b) f-ratio (new/regenerated production) and (c) phytoplankton biomass (mgC m⁻³) in offshore boxes. Each color corresponds to a box (see legend).**





**Figure 11: Spring average (March–May) of vertical phytoplankton transport at the bottom of offshore boxes by (a) advection and (b) diffusion (vertical mixing) and (c-d-e-f) lateral transport at the (c) northern, (d) southern, (e) western and (f) eastern boundaries (defined positive inward, so vertically upward) (molC m⁻² s⁻¹). Each color corresponds to a box (see legend).**





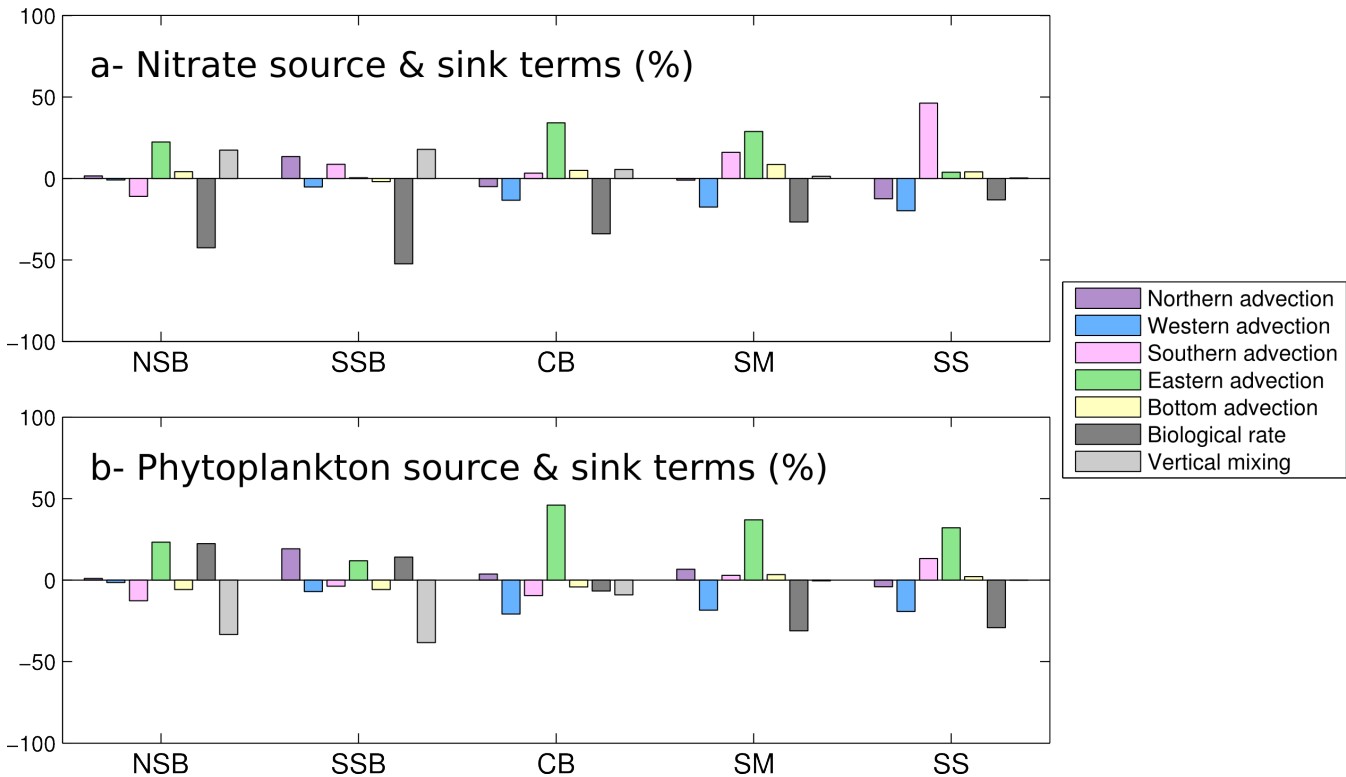

**Figure 12: Spring average (March–May) contribution of the different source and sink terms of (a) nitrate and (b) phytoplankton biomass in offshore boxes ( %, positive inward): northern, western, southern and eastern horizontal advection, vertical advection and diffusion (vertical mixing) at the bottom and the net local biological rate of change.**



**Figure 13: Schematic of the spring average contribution of the different source and sink terms of nitrate and phytoplankton concentration within each box defined in this study. Each color corresponds to a box (see legends in Fig. 2 for coastal and Fig. 8 for offshore boxes). Arrows indicate horizontal advection, circles indicate vertical advection, squares indicate vertical mixing and triangles indicate biological processes. Within a circle or square, a white point indicates a source while a white cross indicates a sink; biological processes are a source/sink if a triangle heads upward/downward, respectively. The size of arrows, circles, squares and triangles indicates the magnitude of the contribution of each source/sink term. For offshore boxes, the information is equivalent to that given in Fig. 12.**





**Figure 14: Seasonal climatology of (a) wind intensity (m s$^{-1}$), (b) upwelling intensity (vertical velocity at the bottom, in m s$^{-1}$), (c) zonal velocities (m s$^{-1}$) and (d) meridional velocities (m s$^{-1}$) within and at the boundaries of coastal boxes.**





Figure 15: (a) Annual average residence time (in days) of upwelled water masses in coastal boxes and (b) spring average (March–May) residence time (in days) of upwelled water masses in offshore boxes.