# Peer review of "What drives the spatial variability of primary productivity and matter fluxes in the North-West African upwelling system ? A modelling approach"

_Biogeosciences, 2016_

## Referee Comment (RC1) · Anonymous Referee #1 · 7 Jun 2016

**Summary**

The paper by Auger et al. has the goal to explain the spatial variability of chlorophyll in the North-West African upwelling system. To do so, Auger et al. present the results of a modelling study based on the coupled models ROMS and PISCES run on a high resolution regional grid. Their results show that the upwelling intensity in the form of vertical velocity does not necessarily follow the forcing seasonality, but is strongly influenced by topography and alongshore flow. They also show that local productivity does not always depend on the local upwelling of nutrients, but it can

rely in great part on the lateral advection of nutrients. In fact, while productivity in the northern and southern sectors of the upwelling system depends on the vertical nitrate supply, the central Cape Blanc subregion is particularly impacted by this convergent lateral advection, both from the northern and the southern boundaries. This central sector is also characterized by high levels of regenerated production (about half of the total production) and subduction of water and tracers resulting from this stream convergence. Auger et al. also show that, except in the Senegalo-Mauritanian region, phytoplankton biomass distribution itself depends on the advection of phytoplankton from the adjacent regions. As regard to the offshore waters, Auger et al. assert that the fast decline of phytoplankton concentration in the region above Cape Blanc is due to strong vertical mixing that redistribute it below 100m, limited offshore advection and strong southward alongshore transport. On the contrary, below Cape Blanc lateral advection of nutrients and phytoplankton sustain a wider offshore extension of the chlorophyll concentration. Around Cape Blanc the flow convergence explains the persistent wide chlorophyll extension.

**Contribution**

The North-West African upwelling system hosts one of the most rich and productive marine ecosystems. Despite this, dedicated studies on this systems have not been as abundant as for other upwelling regions for a long time. In the last few decades many precious data have been collected in this region and several modelling studies have been focusing on this system. Thanks to these efforts, an always more comprehensive understanding of the local ecosystem as well as of the complex pattern of currents is being achieved. The paper by Auger et al. adds some other important pieces of information to this picture and clarifies the mechanisms that lie behind the observed pattern of chlorophyll in the region.

One important novelty and strength of this paper is comparing the alongshore lateral fluxes of nutrients to their upwelling supply. The authors prove that these lateral fluxes are relevant in most sectors of the system, where the upwelling of nitrate is not the dominant supply of nutrients. They also prove that in one particular region lateral advection of nitrate is the fundamental key to explain the observed pattern of chlorophyll that persists despite the lack of upwelling. Similarly, alongshore advection of phytoplankton can account for as much biomass as the local production in some regions. This conclusions represents a change of perspective in the understanding the dynamics of the upwelling system especially in the nearshore region. It may also be an interesting subject of study in other upwelling systems.

**Recommendation**

I find the paper by Auger et al. particularly new, exciting and suitable to be published in Biogeosciences. The paper is well organized, overall nicely written and the results are relevant.
I strongly suggest that it is accepted for publication after minor revisions are done.
Suggested revisions are listed in the next Major and Minor Comments sections.

**Major Comments**

**1)** Verbose "Results" Section
The paper by Auger et al. has the great merit of analyzing in depth the results of the model and of using cross-analysis of several different quantities to validate the hypothesis of the Authors. However, I have found the Results section pretty heavy to read, especially due to the amount of numbers listed within the text. This has a peak in sections 3.1.3 and section 3.2.3. I strongly suggest to summarize the results sections since the plots already contain much of the information that is explained in words in

these chapters. The many many numbers listed in sections 3.1.3 and section 3.2.3 could instead be included directly in the pictures, for example above the bars, or in a table. As a general comment, I suggest to summarize/reorganize the Results section and use it to highlight general features and important trends in the Figures, rather than describing them element by element. This way the readers can really grasp the major highlights and findings without getting lost in too much information, while they can still look at the plots/tables for more details.

**2)** Why Spring
It was not clear to me until the Discussion section 4.3 the reason why some of the analysis in the paper was focusing on spring and I am still not sure that I have comprehended all the rational behind it. I suggest to motivate this choice more in depth before to present the results to the reader, stressing on the motivations that lie behind the choice of presenting a detailed analysis of the fluxes in this specific season only, despite all the known subregional variability of upwelling and seasonality in the system.

**3)** Model Evaluation: Nitrate
Most of the discussion in the paper by Auger et al. is focusing on Nitrate, however there is no model evaluation of the nitrate distribution in the model. I suggest to include this in the paper.

**4)** Box analysis Figures
The paper by Auger et al. presents most of the results in the form of fluxes analysis. However, some of the Figures are confusing.
Figure 7, Figure 13: Why are there 2 arrows of different color and size for each one of the lateral fluxes between the boxes? Eg, in Figure 7 the meridional flux between SS and SM in the nearshore is represented by both a large red arrow and a not so large orange arrow, so there are 2 arrows of different size for a single flux. What does this

mean? Isn't the size of the arrows proportional to the intensity of the fluxes? Is the size of the arrows of one box comparable with the size of the arrows in the other boxes or do each box have a different scale?

**Minor Comments**

1) page 3: title "Material and Methods" maybe should be "Materials and Methods"? (missing s)

2) page 3, lines 24-25: What is the output frequency of the model? Monthly means?

3) page 4, lines 30-31: The large cyclonic recirculation introduced here and fed by the NECC is generally referred to as Mauritanian Current [J. Arístegui et al./Progress in Oceanography 83 (2009) 33–48] in its northward alongshore component. This current is referred to again in page 5 lines 7-12

4) page 5 lines 1-2: "Maximum velocity is found equatorward in the coastal upwelling jet": this sentence is a bit odd as regard to English syntax, it may be reformulated in a clearer way

5) page 5 line 3: (and all the next occurrences) Cape Boujdour (FR) in English is called Cape Bojador

6) page 6 lines 9-18: this block of lines sounds more like a "Model Results" paragraph and it seems out of place; if moved somewhere else the Box Analysis section actually sounds much more coherent; maybe it can be located in a more adequate position

7) page 10 line 12: I don't think that "enlightened" is the right word here (eg, enlightened = Having or showing a rational, modern, and well-informed outlook; spiritually aware)

8) page 10 line 28: at the bottom of the offshore boxes (missing "the" before "offshore boxes")

9) page 10 lines 31-32: the sentence about vertical velocities and nitrate supply is not clear to me, this finding may be better explained

10) page 11 line 10  page 12 line 2: Annual Mean? Or Spring Mean? In general, I suggest always to remark throughout the discussion and conclusions section whether your sentences are referring to the annual mean or spring mean analysis

11) Figure 14: Why are the lines in subplot c and d dashed?

---

## Referee Comment (RC2) · I. Ruvalcaba Baroni (Referee) · 17 Aug 2016

**Summary**

The paper of Auger et al. analyses in detail the North-West African upwelling region through a physical-biogeochemical model and largely contributes to a better understanding of the seasonal variability and spatial distribution of phytoplankton biomass and production for this area. It gives a good description of most processes responsible for transporting nutrients and phytoplankton, as well as an indication of the dominant processes between local phytoplankton growth, lateral transport or regenerated pro-

duction. The used model has the advantage of allowing separation of processes that are normally difficult to detach one from another. For example, it differentiates the effect of lateral advection from that produced by the general ocean circulation or vertical mixing. Furthermore, it allows separation between local new production and lateral phytoplankton transport. When compared to satellite observations, the model gives an accurate surface ocean circulation and chlorophyll distribution, which are used to validate the model. Interestingly, they find that nutrient upwelling, despite being significant in the entire upwelling system, is not always the dominant process for sustaining primary productivity. Instead, lateral transport of nutrients and phytoplankton appear to be the dominant process in most of the studied subregions, with an important southward-alongshore and cross-shore advection off the Saharan Bank and south of Cape Blanc, respectively, as well as an intrusion of nutrient-rich waters from the offshore Guinean upwelling to the Senegalo-Mauritanian subregion.

**Recommendation**

This paper adds important information about relevant mechanisms influencing primary production in the North-West African upwelling region, which is one of the most productive marine ecosystem, but also poorly understood. In this regard, the paper of Auger et al. gives new insights on how and why primary productivity varies temporally and spatially in this region. The paper is well written, with clear figures and well organized. Therefore, I highly recommend it for publication in Biogeosciences, after minor corrections.

**General comments**

This paper has a very complete model analysis of all major processes affecting primary productivity and nutrient transport. The only major critique I have is that the authors should have made more comparisons between their model results and actual observations or other model results. I realize that there is not so much published data from this region, however, from this paper it is not always clear, especially in the discussion

section, what is known, what is new, what was previously suggested and what was confirmed by their model results. Nevertheless, with some rephrasing and addition of visual comparison where possible, this should be easy to improve.

**Major and specific comments**

1) Figure 1: Why not add a figure comparing model and measured data for nitrate in, for example, surface waters? Besides chlorophyll and surface currents, there is not much comparison with observations, which would add robustness to the model results.

2) Section 3.1.1, lines 15-16: The sentences "Offshore transport of nitrate... cross-shore velocities" do not really fit this paragraph because they describe the nutrient behavior while the paragraph is about ocean circulation. They should be moved to line 23, before "Vertical (upwelling-induced) nitrate supply...". Then, remove "(see above)" and add (Fig. 3d) at the end of "cross-shore velocities".

3) I think it will be nice to indicate somewhere (for example in section 3.1.1, end of line 6) that negative velocities indicate outside transport (out of the box) while positive values indicate inward transport (inside the box). Am I correct? If this is the case, what does negative advection mean in bottom waters?

4) Section 3.1.2 (page 7 and 8): This section is a bit difficult to follow due to its structure. Why not start this paragraph with the description of Figures 3e-h and then introduce Figures 4 and 5? So mainly move "Off Cape Blanc, phytoplankton biomass....(Fig. 3d)." to the beginning of the paragraph. Then continue with "Phytoplankton biomass, averaged over ... (Fig. 3a)." and then focus on PP. In this case, I would plot figure 4c (phyto) in position 4a, figure 4a (PP) in position 4b and figure 4b (f-ratio) in position 4c. For consistency, I would also rename the title of the section as: "Phytoplankton biomass, primary production and phytoplankton fluxes". Perhaps. the paragraph will then need some rephrasing.

5) Figures 7 and 13 are very similar and it takes some time to understand their main

differences, as well as why both are relevant. It will be good to stress out more the differences and mention why both are relevant to this work. It is also not very clear why the spring average is now more relevant than the annual average, and why spring more than other seasons (i.e. besides being the most productive, what are the advantages to show spring processes?).

Is there a way to combine these 2 figures and their respective descriptions in a summary section?

6) Section 3.2.1, line 32: What does "vertical velocities are pointing downward" mean in this context?

7) Section 3.2.2, lines 20-25: Is there any other model or measurement data also showing the strong diffusion in this region? It would be nice to compare these results to published data or at least mention how reliable these results are.

8) Section 4.1, lines 10-16 (page 14): Same as comment 7. Is there any model or measurement data to compare these results to?

9) Section 4.3, line 18 (page 17): Are the results of intrusion of nutrient-rich waters in the Senegalo-Mauritanian region in agreement with Lazaro et al., 2005 and Mittel-staedt, 1991? If so, it will be good to specify it: "originating from the Guinean upwelling. This is in agreement with previous finding of Lazaro et al., 2005 and Mittelstaedt, 1991." How do the results of Lazaro et al., 2005 and Mittelstaedt, 1991 compare to those of this paper?

10) Figure 14: Could these model parameters be compared to measurements, other model results or a theoretical average?

11) Section 3.2.2 starts with "Annual mean PP", but it should be "Spring average PP ...", I think.

12) Section "Discussion", line 11: This comment relates to point 5). Because it is not really mentioned before, I would shortly indicate, at the end of this paragraph, why

spring processes are analyzed.

13) Section 4.1, line 22 (page 13): "Therefore it gives us confidence..." how much confidence? Some, good, very good...? I would indicate it here.

14) Section 4.1, line 23 (page 13): "For this latter purpose, we further ..." This sentence does not really explain in what way the analysis of seasonal cycles at the edge of the coastal boxes will add confidence to the model. Especially because these sensitivity results are not compared to observations or theoretical behavior. Please be more specific about the importance of this analysis.

15) Section 4.2, lines 33 (Page 14) and line 1 (page 15): Are these new results or do they include those of Barton et al., 2004, García-Muñoz et al., 2004 and Karakaş et al., 2006? Please specify which part of these results are new or used to confirm previous ones.

16) Section 4.2, line 15 (page 15): "Zooplankton excretion..." What is the explanation of the zooplankton excretion being more important for regenerated production in areas with lateral transport?

17) Section 4.3, lines 17-19 (page 15): "This supports the idea that, ..." Is this sentence in agreement or in opposition to the previous hypothesis? What do other authors say about lateral transport of organic matter in this area?

18) Section 4.3, lines 14-19 (page 16): "For this purpose, we focus...", this sentence should be briefly mentioned before as mentioned in comment 12, and here, it should be improve. Please clarify why this specific setting (maximal coastal upwelling at the Saharan Bank and maximum phytoplankton extension off Cape Blanc) is a good scenario to be tested with the model.

**Minor comments/Technical corrections**

1) Section"Abstract" line 29: Did you mean "lateral advection transports coastal nutrients..."?

2) Section "Introduction" line 2: "... and seasonally variable one...". I would remove "one".

3) Section "Introduction" line 29: There is a missing comma, "Here, we ...",

4) Subsection "Model validation" line 22: Are AVHRR initials? If so, they probably should be written in full, here.

5) Subsection "Model validation" line 25: To ease the understanding of Figure 1, I would replace the titles "SeaWiFS/Drifters SVP" and "ROMS" by "Observations" and "Model", respectively. I could not figure it out what SVP means. Then, add in caption "from observation given by SeaWiFS satellite data... Same for ROM-PISCES model in..."

6) Subsection "Model validation" line 6 and 7 (page 5): I would remove "see" from both "see Fig. 1" in this paragraph. I would also move the second fig. reference, line 7, to the end of the sentence: "both in the model and in the data during summer (Fig. 1)". This is, because the NECC is not entirely shown in the figure and creates some confusion if the figure reference is placed after mentioning the NECC.

7a) Subsection "Model validation" line 14 and Figure 1: If maximum values of chlorophyll in both satellite and model data go up to 10 mgChl m$^{-3}$, why stop at 5 mgChl m$^{-3}$ in the color bar of Fig. 1?

7b) Figure 1: At first I could not see the 5 coastal areas. Maybe by coloring the lines in white, instead of black, the boxes will be more visible. In the caption of Figure 1, "The ten boxes" could also be replaced by "The 5 coastal and 5 offshore boxes", so it is even more clear that the 5 coastal boxes are shown in the figure.

7c) Missing figure: Please remove the sentences "Main surface currents and deep water masses over ... South Atlantic Central Water" or add the missing plot and describe it in the main text, but only if it adds relevant information to the article.

8a) Section 3.1, title: To simplify, why not just call it "Meridional variability in the coastal region"?

8b) Section 3.1, title: Maybe add "Annual average of the meridional variability in the coastal region"

9) Section 3.1.1, line 7: "on the edge of coastal boxes" gives a vague description of what we are actually looking at in the figure. Maybe replace by "on each edge of the coastal boxes (i.e. North, South, West and bottom)".

10a) Figure 2: Are the "BOX" and the black lines needed for figure 2a and 2b? I think they do not add much information, unlike for Fig. 2c, 2d, 2e, and 2f. These plots (Fig. 2a and 2b) alone clearly represent the entire box.

10b)Figures 2-3: These figures are good, but they are a bit difficult to get at first (especially Fig. 2, since it is the first one of the series). To make Fig. 2 easier to understand, I would separate the main title "Wind and currents" to "Wind (m s$^{-1}$)" and "Velocities (m s$^{-1}$)". This will make it clear that this figure is showing 2 different things. I would write "Wind (m s$^{-1}$)" on top of Figure 2a and 2b and "Velocities (m s$^{-1}$)" on top of Figures 2b, 2c, 2d, 2e and 2f, as done for Fig. 3. Please use velocities instead of currents, because the word current is not as often used as velocity in the main text. Please also change the title of figure 2c to "c-Bottom (upwelling)" or "c-Bottom (vertical velocities)". The chosen title, "c-Bottom velocity" is misleading, as it suggests horizontal bottom velocities.

10c) Please check the sub-numbering of the figures, some do not follow a logic order.

11) Section 3.1.2, line 30: Please rephrase "... of PP in the boxes north and south of Cape Blanc are simulated, respectively."

12) Section 3.1.3, lines 26-30: This sentence is difficult to follow, maybe rephrase to "... the total rate of change of nitrate concentration and phytoplankton biomass in each coastal box are presented..."

13) Section 3.1.3, line 8: replace "is not anymore" by "is no longer"

14) Section 3.1.3, line 13: rephrase "from the northern Saharan Bank and the Senegal-Mauritanian region, respectively."

15) Section 3.1.3, line 31 (page 9): remove "tentative"

16a) Section 3.2, title: Same as comment 8: To simplify, change it to "Meridional variability in the offshore region".

16b) I think, here, the title of section 3.2 should also mention that the results are now only for spring: "Spring meridional variability in the offshore region". For me this was not clear until the discussion.

17) Section 3.2.2, line 27: I would remove "finally".

18) Section 4, line 10: Please replace "we will seek to explicit" by "we explain"

19a) Section 4.3, line 10 (page 17): Space missing, please correct "that filaments"

19b) Section 4.3, line 14(page 17): Please replace "hypothesis" by "processes". The points mentioned before are not hypotheses.

20) Figure 8: Same comment as 10a for Figure 2 regarding the black lines and the BOX.

21) Figure 11a and 11b are missing.

22) Figure 14: Why Fig 14c is suddenly represented with dashed lines? What do the plain and dashed lines represent in Fig. 14d?

23) Please check the usage of acronyms, especially in the legend of the figures. I would always use the full subregion names or always the acronyms. Personally, I would avoid the acronyms (also in the text).

24) Section 4.3, line 8: Please correct "2ZOLTLathuiliere2008"

25) I would replace most "see Fig." by "Fig.", except maybe the one in line 24, page 4.

**Further suggestions**

1) Section 4.1, line 15-16 (page 13): Rephrase "It appears that coastal upwelling" by "However, coastal upwelling". This makes it more clear that this information comes from the literature and not the model results.

2) Section 4.1, line 20 (page 13): Replace "The simulated" by "Our simulated"

3) Section 4.2, line 4: "for 50 % of new production", please indicate if it is a model result or a general statement. Can this be seen in one of the figures? If yes, please indicate in which one.

4) Section 4.2, lines 6-8: Would it be possible to compare this with observations of mixed layer depth (MLD) off Cape Blanc? If the MLD is large there, it will give some extra validation for this result.

5) Section 4.2, lines 9-10: I would replace "Usually" by "In general", and in the following sentence, I would add "However, in our results, the water ...."

6) Section 4.2, line 35 (page 15): The word "provide" is maybe not the best word, here. I suggest "lead to", instead.

7) Section 4.2, lines 1-2 (page 16): To ease the reading, I would briefly re-mention why this is the case, instead of referring to the previous section. Section 4.1 mentions several points and it is not exactly clear to which one this sentence refers to. I would also remove "(see above)", it is not clear to what it is referring to.

8) Section 4.3, line 27: Please add "In our results, the vertical mixing..." and replace "Indeed" by "This is even more visible in the results for spring, where the mixed layer...".

9) Section 4.3, lines 33-34: To better differentiate between the new results and published data, please rephrase the sentence as follows: " In fact, the vertical mixing, as
previously suggested by Huntsman and Barber 1977, is also responsible for ..."

10) Section 4.3, line 1: Maybe replace "participate to" by "partly".

11) Section 4.3, line 7: To help the reader, it would be nice to briefly mention what is the hypothesis of Lathuilère et al., 2008.

12) Section 4.3, lines 8-9: To better differentiate between the new results and published data, please add " In our results, the advection by ..." and "...off Cape Blanc, in agreement with Kostianoy and Zatspein, 1996.

13) Section 4.3, line 10: To better differentiate between the new results and published data, please add "may enhanced cross shelf transport, as also shown by satellite data in Lathuilère et al., 2008".

---

## Referee Comment (RC3) · Anonymous Referee #3 · 30 Aug 2016

Summary In this study a coupled physical-biogeochemical model is used to understand the spatial variability of primary production in the NW African upwelling system. Homogeneous regions are chosen based on similar horizontal current patterns and individually analysed with regard to drivers of primary productivity. Their results indicate that coastal upwelling is modulated by coastal topography and large-scale geostrophic currents, effectively enhancing upwelling off Cape Boujdour and limiting it off Cape Blanc. Coastal upwelling of nitrate, and therefore new production, was found to be significant everywhere except Cape Blanc, but dominant only in the northern Saharan Bank and Senegal-Mauritanian regions, elsewhere it is predominantly provided by meridional advection. Therefore, off Cape Blanc the net coastal phytoplankton growth is sustained by high levels of regenerated production. The offshore pattern of nitrates and phytoplankton is driven by coastal circulation patterns: in the north of the domain the coastal circulation is predominantly alongshore explaining the limited offshore extent of high productivity and further south, large filaments associated with large offshore fluxes result in high productivity much further offshore.

Recommendation This work is beneficial to the understanding of physical-biological interactions in the NW African upwelling system and should be published, with some minor corrections and condensing of some of the sections.

General comments Title could be shortened to: What drives the spatial variability of primary productivity and matter fluxes in the North-West African upwelling system? A modelling approach.

Throughout the manuscript 'explicit' is used as a verb. It is not a verb, it can be used as a noun (eg. The explicitness of the data allow us to draw some very solid conclusions) but usually an adverb (The data allow us to explicitly show that...) or adjectve (e.g. The data is explicit, it shows that...).

It seems a bit unclear as to why the spring means are used for the offshore domains and annual means for the coastal domains. It is mentioned only later in the discussion, but it should be clearer sooner. Why not show the spring mean for coastal and offshore domains (surely using the annual mean for the coastal domain masks the seasonal signal and is therefore unrealistic?). Does it make sense to link the coastal and offshore domains in terms of the offshore fluxes for example if you are looking at averages for different periods?

The results section is very laborious and therefore difficult to read, especially sections 3.1.3 and 3.2.3 (the percentages in paretheses are not necessary), and could be shortened.

[Figure]

Specific comments page 2, line 4: 'of coastal topography are' should be 'of coastal topography is'

page 2, line 5: '...and then the response of nutrient upwelling to wind forcings'. This is unclear. Are you saying that the large scale circulation pattern impacts the wind driven upwelling of nutrients?

Introduction Page 3, line 17: 'in regards of enironmental forcings' should be 'with regard to enironmental forcings'

Page 4, line 2: 'To this end, comparative box analysis...' should be 'To this end, a comparative box analysis...' or 'To this end, comparative box analyses......have been conducted'.

Page 4, line 3, 'Those subregions' should be 'The subregions'

Page 4, line 4,' in regards of' should be 'with regard to'

Page 4, Line 14: 'explicit' cannot be used as a verb, try 'identify',

Last two paragraphs of Introduction are laborious and could be more succinctly summarised.

Last two sentences of Introduction seem out of place.

Methods Page 4,line 26: unbalanced parenthesis

Page 5, line 23: 'thinner' is ambiguous in this context, 'narrower' is clearer

Page 6, line 3: 'The upwelling filaments off Cape Ghir and Cape Boujdour are responsible for strong seaward deflections of the coastal current.' I wouldn't necessarily say that the filaments are responsible for the seaward deflection – they are 'connected', both associated with the same initial mechanism (perhaps wind/topography) and then they probably enhance one another.

Page 6, Line 2: 'explicit' cannot be used as a verb, try 'identify'

Page 6, line 2-4: '. . ........the meridional variability of primary productivity off the NW African coast, we carried out a box analysis focusing on nitrate (the main limiting nutrient) and phytoplankton carbon budgets (12–27° N, see Fig. 1)',

rather say:

'. . ........the meridional variability of primary productivity off the NW African coast between 12–27° N, we carried out a box analysis focusing on nitrate (the main limiting nutrient) and phytoplankton carbon budgets'

Page 6, line 7: '...was split ito five latitudinal bands.' rather: '...was split into five latitudinal bands (see Fig. 1).'

Page 6, line 12: remove 'On the opposite,', start with 'In the southernmost...'

Page 6, line 17: what do you mean by 'globally'? It usually refers to something involving the whole globe/world.

Page 6, line 20: 'In like manner...' rather 'Similarly..'

Results Page 8, line 6: 'Wind curl shows a clear maximum off Cape Blanc but a weak meridinal variability'. This sentence is not clear. Do you mean that the meridional variability in wind stress curl is weak or do you mean that the alongshore variability of meridional wind stress is weak?

Figure 2 c : this is labelled as upwelling intensity (vertical velocity at the bottom). For upwelling intensity, it would be better to use vertical velocity at the base of the Ekman layer (your 100 m depth of the boxes is probably too deep?).

Page 8, line 24: remove 'inversely'

Page 8, line 32: '...associated to...' should be '...associated with...'

Page 9, line 11: 'does not translate in...' should be 'does not translate into....'

Page 9, line 14: 'Noteworthy, the phytoplankton biomass is found maximum off Cape

[Figure]

Blanc and the South Saharan Bank contrasting with minimum upwelling-induced nitrate supplies (Fig. 3a).

rather:

'It is noteworthy that maximum phytoplankton biomass is found off Cape Blanc and the South Saharan Bank despite the fact that upwellling-induced nitrate supplies are at a minimum at those locations (Fig. 3a).'

Section 3.1.3: laborious

page 10, line 6: 'sinks' should be 'sink'

page 11, line 12: 'enlightened' is very archaic in this context. Replacing it with 'euphotic zone' would be better.

Page 11, line 14: 'explicit' cannot be used as a verb and whole sentence is unclear.

Page 11, line 17: 'Alternatively' since you're not offering an alternative to a previous statement, something like 'On the other hand' works better.

Page 11, line 22: 'Noteworthy, at the western...', change to 'It is noteworthy that at western boundaries velocities are...'

Page 11, line 23: replace 'happen to be' with 'are'

Page 11, line 29: replace '..falls in the same order of magnitude than diffusion...' with '..is the same order of magnitude as diffusion...'

Page 11, line 31: replace 'Noteworthy...' with 'It is noteworthy that vertical nitrate supply...'

Page 12, line 10: In the text it states that Fig. 10 is annual mean, but the figure caption says Spring mean.

Page 12, line 18: replace '.…...., in less manner,....' with '. . ., less so,...'

Page 13, line 2: In the text it states that fig 12 shows the annual mean source and sink terms but the figure caption says it is the spring mean.

Discussion Page 14, line 9: 'in relation with the...', should be 'in relation to the...'

Page 14, line 10: 'Finally, we will seek to explicit...' should be 'Finally, we will seek to identify...'

Page 14, line 13: 'In our simulation, the meridional variability of coastal upwelling is not correlated to the local variability of wind-driven Ekman transport and Ekman pumping. This result questions the estimation of vertical velocities based on local wind forcing that were commonly used in EBUS'.

- two points on this statement: 'were' should be 'are'

The estimation of upwelling using alongshore wind stress is for vertical velcoities at the base of the Ekman layer. Your level of 100m, or the bottom in places shallower than 100 m, may be too deep.

Page 14, line 16: You state that the large scale transport could be a factor explaining the mismatch in upwelling intensity and Ekman transport. With the model output you can calculate it directly to verify your statement.

Page 14,line 22: '...explicit...', can't be used as a verb. You could use 'identify'

Page 14, line 23 and figure 14: you use the bottom velocity to assess the sensitivity of coastal upwelling to wind forcing. You should rather use vertical velocity at the base of the Ekman layer.

Page 14, line 26: 'lead' should be 'leads'

Page 15, line 18: instead of 'Albeit' use 'Although'.

Page 16, line 31: sentence starting with 'This points a gap...' is confusing

Page 17, line 13: 'the coast, the wind stress curl...' should be 'the coast, (ii) the wind

stress curl...' Page 17, line 14: 'hypothesis' should be hypotheses'

Page 17, line 15: 'explicit' should be 'identify'

Page 17, line 20: 'associated to' should be 'associated with'

Page 17, line 22: 'Our results indicate downward and upward wind-induced Ekman pumping of respectively north and south of Cape Blanc' should be 'Our results indicate downward and upward wind-induced Ekman pumping north and south of Cape Blanc respectively'

Page 18, line 1: 'participate' should be 'help'

Page 18, line 8: '2ZOTLathuliere2008' – a latex referencing bug?

Page 18, line 10: 'thatfilaments' should be 'filaments'

Conclusion Page 18, line 21: 'of the primary production spatial distribution in' should be 'of the spatial distribution of primary production'

Page 18, line 25: ' production in' should be 'production with'

Page 18, line 31: 'excepted' should be 'except'

Figures Figure 1: include the box labels that you use in the text and in other figures

Figures 1-4: in some you include just the abbreviations of the box areas, in others you have the full name. When you don't have the full names in the legend, you could include them in the caption

Figure 6: label x-axis with latitude as well, or at least show where north is

Figure 10 and 12: the captions dont agree with the text (Annual vs. Spring mean)

Figure 14: it is not clear how these averages are calculated (in the caption or in the text). In the caption you state: 'within and at the boundaries of coastal boxes'. Is it an average of meridional wind, bottom velocity, cross-shore velocity and alongshore

velocity within the entire coastal strip? If so, does this make sense, given that your interest is the meridinal variability of primary productivity.

Please also note the supplement to this comment:
http://www.biogeosciences-discuss.net/bg-2016-156/bg-2016-156-RC3-supplement.pdf

————————————————————

---

## Author Comment (AC1) · 30 Sep 2016

We wish to thank referee #1 for his/her detailed analysis and his/her thoughtful comments, which will improve the quality of this manuscript. Here, you will find a detailed reply to each comments :

Response to Referee#1's Comments

**Major Comments**

**1) Verbose "Results" Section**

The paper by Auger et al. has the great merit of analyzing in depth the results of the model and of using cross-analysis of several different quantities to validate the hypothesis of the Authors. However, I have found the Results section pretty heavy to read, especially due to the amount of numbers listed within the text. This has a peak in sections 3.1.3 and section 3.2.3. I strongly suggest to summarize the results sections since the plots already contain much of the information that is explained in words in these chapters. The many many numbers listed in sections 3.1.3 and section 3.2.3 could instead be included directly in the pictures, for example above the bars, or in a table. As a general comment, I suggest to summarize/reorganize the Results section and use it to highlight general features and important trends in the Figures, rather than describing them element by element. This way the readers can really grasp the major highlights and findings without getting lost in too much information, while they can still look at the plots/tables for more details.

**We agree that some parts (in particular section 3.1.3 and 3.2.3) of the « Results » Section is dense, and that our major findings are perhaps lost in too much numbers listed in the text. According to the referee's suggestion (shared with referee #3), numbers in sections 3.1.3 and 3.2.3 will be removed from the text. The Results Section will also be clarified to higlight the major features of the study region shown in the figures.**

**2) Why Spring**

It was not clear to me until the Discussion section 4.3 the reason why some of the analysis in the paper was focusing on spring and I am still not sure that I have comprehended all the rational behind it. I suggest to motivate this choice more in depth before to present the results to the reader, stressing on the motivations that lie behind the choice of presenting a detailed analysis of the fluxes in this specific season only, despite all the known subregional variability of upwelling and seasonality in the system.

**Indeed, the three referees noted that the justification of why we partly focus on spring should be done earlier than in section 4.3. The main justification is that observed offshore extension of Chl-a do present a marked seasonal variability with a peak in boreal spring. Therefore, focusing only on annual averages would have raised questions about the significance of our results during the time period that sees most of the offshore export. Choice has thus been made to show annual average but also the spring period.**

**According to shared referees' comments on this point, the choice of the spring season for the analysis of offshore boxes will then be clearly motivated in the introduction by modifying the sentence p.3/l.8 from :**

**« The following section is focused on the description of the meridional**

variability of annual wind forcings, ocean response and primary productivity as simulated by the model in the different coastal boxes (Section 3.1) and offshore boxes (Section 3.2). »
to
« Then, we describe the meridional variability of wind forcings, ocean response and primary productivity as simulated by the model in the different coastal (Section 3.1) and offshore boxes (Section 3.2), on annual mean and also during spring (seasonal maximum of the chlorophyll offshore extension as shown in Lathuilière et al, 2008). »

This will be also mentionned at the beginning of the Results Section 3.2.1 modifying p.10/l.9 from :
« The offshore extension of chlorophyll has been shown to display a marked seasonal variability with a maximum in spring (Lathuilière et al., 2008; see Fig.1). In the offshore region, the chlorophyll variability depends on the export of coastal productivity. Additionally, the wind stress can be responsible for vertical mixing that enhances the exchanges of inorganic and organic matter between the euphotic and aphotic layers. The vertical nutrient supply to the enlightened surface layer and the phytoplankton export below the euphotic layer may also be enhanced by positive/negative Ekman pumping, respectively linked to positive/negative wind stress curl. In order to explicit the offshore extension in spring of the rich phytoplankton pattern, mean wind forcings from March to May (i.e. wind intensity and wind curl) are first presented in Fig. 8 (a & b). During these months, the wind intensity increases from the northern Saharan Bank to Cape Blanc (where it peaks) and then decreases southward (Fig. 8a). »
to
« In the offshore region, the chlorophyll seasonal variability may depend on the export of coastal productivity. Additionally, the wind stress can be responsible for vertical mixing that enhances the exchanges of inorganic and organic matter between the euphotic and aphotic layers. The vertical nutrient supply to the euphotic surface layer and the phytoplankton export below the euphotic layer may also be enhanced by positive/negative Ekman pumping, respectively linked to positive/negative wind stress curl.
Off NW Africa, the offshore extension of coastal chlorophyll has been shown to display a marked seasonal variability with a maximum in spring (Lathuilière et al., 2008; see Fig.1). Thus, spring averages from March to May were considered to investigate the factors driving primary productivity in offshore boxes. Mean spring wind forcings (i.e. wind intensity and wind curl) are first presented in Fig. 8 (a & b). During spring, the wind intensity increases from the northern Saharan Bank to Cape Blanc (where it peaks) and then decreases southward (Fig. 8a). »

3) Model Evaluation: Nitrate
Most of the discussion in the paper by Auger et al. is focusing on Nitrate, however there is no model evaluation of the nitrate distribution in the model. I suggest to include this in the paper.
We propose to add, in Figure 1, white contours of nitrate concentrations at 100m depth (the depth of the boxes defined for our

**analysis) both in the model and the CARS 2009 global atlas product. This will permit us to show the sharp change in nutrient concentrations which occurs off Cape Blanc between nutrient-poor North-Atlantic Central Water north of Cape Blanc, and nutrient-rich South-Atlantic Central Water south of Cape Blanc. This will be mentioned in the text p.5/l.12 :**
**« Noticeably, the flow of the undercurrent over the slope is always poleward (not shown) in agreement with observations (Mittelstaedt, 1983). Besides the model accurately represents, at latitudes around Cape Blanc, the sharp gradient of nutrient concentrations in upwelling source waters between nutrient-poor North Atlantic Central Water (NACW) and nutrient-rich South-Atlantic Central Water (SACW), respectively north and south of Cape Blanc (see contours in Fig. 1). This actually results from the deepening of the poleward undercurrent transporting SACW and its intensive mixing with NACW north of Cape Blanc (Mittelstaedt, 1983). »**

4) Box analysis Figures
The paper by Auger et al. presents most of the results in the form of fluxes analysis. However, some of the Figures are confusing. Figure 7, Figure 13: Why are there 2 arrows of different color and size for each one of the lateral fluxes between the boxes? Eg, in Figure 7 the meridional flux between SS and SM in the nearshore is represented by both a large red arrow and a not so large orange arrow, so there are 2 arrows of different size for a single flux. What does this mean? Isn't the size of the arrows proportional to the intensity of the fluxes? Is the size of the arrows of one box comparable with the size of the arrows in the other boxes or do each box have a different scale?
**As mentioned in their captions, Figures 7 and 13 present the « contribution of the different source and sink terms of nitrate and phytoplankton concentration within each box defined in this study. Each color corresponds to a box ». The size of the arrows of one box is then comparable with the size of the arrows of other boxes in terms of their contribution to the nitrate or phytoplankton concentration in each box. As this is only mentioned in the figure captions, we propose to also mention it in the text when these figures 7 and 13 are presented at the end of Section 3.1.3 and 3.2.3, respectively.**

Minor Comments

1) page 3: title "Material and Methods" maybe should be "Materials and Methods"? (missing s)
**We agree and will take into consideration this suggestion.**

2) page 3, lines 24-25: What is the output frequency of the model? Monthly means?
**Model outputs are saved as 5-day averages.  This information will be added in the caption of Figure 1 by replacing the sentence :**
**« Same for ROMS-PISCES in (c) winter and (d) summer. »**

**by « Same seasonal climatology computed with the 5-days outputs of ROMS-PISCES in (c) winter and (d) summer. »**

3) page 4, lines 30-31: The large cyclonic recirculation introduced here and fed by the NECC is generally referred to as Mauritanian Current [J. Arístegui et al./Progress in Oceanography 83 (2009) 33–48] in its northward alongshore component. This current is referred to again in page 5 lines 7-12

**We agree and will take into consideration this suggestion by modifying the text p4/l.30 from : « South of 19° N, a large cyclonic recirculation is found between the south-westward flowing Canary Current and the coast, especially in summer when trade winds extend farther north (see Barton et al., 1998; Mittelstaedt, 1983, 1991). »**
**to**
**« South of 19° N, a large cyclonic recirculation is found between the south-westward flowing Canary Current and the coast, especially in summer when trade winds extend farther north (see Barton et al., 1998; Mittelstaedt, 1983, 1991). It generates a poleward alongshore flow at its eastern flank generally referred as Mauritanian Current (Aristegui et al., 2009). »**

**The text will be also modified p.5/l.7 from : « Alternatively, a moderate poleward current (which can be seen as an extension of the NECC, see Fig. 1) lays south of Cape Blanc both in the model and in the data during summer when upwelling-favourable winds are weak. »**
**to**
**« Alternatively, a moderate expression of the poleward Mauritanian Current lays south of Cape Blanc both in the model and in the data during summer when upwelling-favourable winds are weak. »**

4) page 5 lines 1-2: "Maximum velocity is found equatorward in the coastal upwelling jet": this sentence is a bit odd as regard to English syntax, it may be reformulated in a clearer way
**This will be reformulated.**

5) page 5 line 3: (and all the next occurrences) Cape Boujdour (FR) in English is called Cape Bojador
**We agree and will take into consideration this suggestion.**

6) page 6 lines 9-18: this block of lines sounds more like a "Model Results" paragraph and it seems out of place; if moved somewhere else the Box Analysis section actually sounds much more coherent; maybe it can be located in a more adequate position
**The structuration of the manuscript and the way our results are presented relies on the definition of spatial boxes which represent homogeneous subregions in terms of physical-biogeochemical characteristics. Consequently, this is a key element of methodology that needs to appear in the dedicated Subsection « Box analysis » of the Section « Materials and Methods ».**
**We would thus prefer to keep subsection as it is.**

7) page 10 line 12: I don't think that "enlightened" is the right word here (eg, enlightened = Having or showing a rational, modern, and well-informed outlook; spiritually aware)
**« enligthened » will be replaced by « euphotic ».**

8) page 10 line 28: at the bottom of the offshore boxes (missing "the" before "offshore boxes")
**We agree and will take into consideration this suggestion.**

9) page 10 lines 31-32: the sentence about vertical velocities and nitrate supply is not clear to me, this finding may be better explained
**This sentence will be modified as follows : « The vertical nitrate supply by advection off the northern and southern Saharan Bank is particularly weak (inward nitrate transport despite averaged outward velocities due to episodic inward events) in comparison to vertical diffusion. »**

10) page 11 line 10 page 12 line 2: Annual Mean? Or Spring Mean? In general,suggest always to remark throughout the discussion and conclusions section whether your sentences are referring to the annual mean or spring mean analysis
**We agree and will take into consideration this suggestion.**

11) Figure 14: Why are the lines in subplot c and d dashed?
**The lines in (c) will be set solid and the caption of Figure 14 will be modified as follows to give the signification of solid and dashed lines : « Figure 14: Seasonal climatology of (a) wind intensity (negative is upwelling-favourable, m s$^{-1}$), (b) bottom vertical velocity (m s$^{-1}$), (c) zonal velocities (m s$^{-1}$) and (d) meridional velocities (m s$^{-1}$) averaged within and over each edge of the coastal boxes (i.e. North, South, West and bottom ; defined positive inward, so vertically upward), respectively. Each color corresponds to a box (see legends in Fig. 2). In (d), a solid (dashed) line represents a velocity at a northern (southern) edge of a box, respectively. »**

---

## Author Comment (AC2) · 30 Sep 2016

We wish to thank referee #2 for his/her detailed analysis and his/her thoughtful comments, which will improve the quality of this manuscript. Here, you will find a detailed reply to each comments :

Response to Referee#2's Comments

*Major and specific comments*
1) Figure 1:  Why not add a figure comparing model and measured data for nitrate in, for example, surface waters?  Besides chlorophyll and surface currents, there is not much comparison with observations, which would add robustness to the model results.
**In response to referee #1 and #2, we propose to add, in Figure 1, white contours of nitrate concentrations at 100m depth (the depth of the boxes defined for our analysis) both in the model and the CARS 2009 global atlas product. This will permit us to show the sharp change in nutrient concentrations which occurs off Cape Blanc between nutrient-poor North-Atlantic Central Water north of Cape Blanc, and nutrient-rich South-Atlantic Central Water south of Cape Blanc. This will be mentioned in the text p.5/l.12 :**
**« Noticeably, the flow of the undercurrent over the slope is always poleward (not shown) in agreement with observations (Mittelstaedt, 1983). Besides the model accurately represents, at latitudes around Cape Blanc, the sharp gradient of nutrient concentrations in upwelling source waters between nutrient-poor North Atlantic Central Water (NACW) and nutrient-rich South-Atlantic Central Water (SACW), respectively north and south of Cape Blanc (see contours in Fig. 1). This actually results from the deepening of the poleward undercurrent transporting SACW and its intensive mixing with NACW north of Cape Blanc (Mittelstaedt, 1983). »**

2) Section 3.1.1, lines 15-16:  The sentences "Offshore transport of nitrate... cross-shore  velocities" do not really fit this paragraph because they describe  the  nutrient behavior while the paragraph is about ocean circulation. They should be moved to line 23, before "Vertical (upwelling-induced) nitrate supply...".  Then, remove "(see above)" and add (Fig. 3d) at the end of "cross-shore velocities".
**We agree and will take into consideration this suggestion.**

3) I think it will be nice to indicate somewhere (for example in section 3.1.1, end of line 6) that negative velocities indicate outside transport (out of the box) while positive values indicate inward transport (inside the box). Am I correct? If this is the case, what does negative advection mean in bottom waters?
**Indeed, it is mentioned in the figure captions, velocities and tracer fluxes are « defined positive inward, so vertically upward » the boxes, but we will add  this clarification in the text at the beggining of Section 3.1.1.**
**Moreover, « Bottom velocity » will be replaced by « bottom vertical velocity » as suggested by the referee in Minor Comment 10b.**

4) Section 3.1.2 (page 7 and 8): This section is a bit difficult to follow due to its structure. Why not start this paragraph with the description of Figures 3e-h and

then introduce Figures 4 and 5? So mainly move "Off Cape Blanc, phytoplankton biomass....(Fig. 3d)." to the beginning of the paragraph. Then continue with "Phytoplankton biomass, averaged over ... (Fig. 3a)." and then focus on PP. In this case, I would plot figure 4c (phyto) in position 4a, figure 4a (PP) in position 4b and figure 4b (f-ratio) in position 4c. For consistency, I would also rename the title of the section as: "Phytoplankton biomass, primary production and phytoplankton fluxes". Perhaps. the paragraph will then need some rephrasing.

**We do not agree with the reorganization proposed by the referee because we find more logical to go from primary production to phytoplankton biomass, than the contrary.**

5) Figures 7 and 13 are very similar and it takes some time to understand their main differences, as well as why both are relevant. It will be good to stress out more the differences and mention why both are relevant to this work. It is also not very clear why the spring average is now more relevant than the annual average, and why spring more than other seasons (i.e. besides being the most productive, what are the advantages to show spring processes?). Is there a way to combine these 2 figures and their respective descriptions in a summary section?

**This comment is related to comments of the 2 other referees who asked for introducing the choice of the spring period earlier in the paper.**
**The main justification is that observed offshore extension of Chl-a do present a marked seasonal variability with a peak in boreal spring. Therefore, focusing only on annual averages would have raised questions about the significance of our results during the time period that sees most of the offshore export. Choice has thus been made to show annual average but also the spring period.**
**Interestingly, there is very little change in the repartition of the fluxes driving nitrate and phytoplankton concentrations in spring compared to annual average, as shown by the little difference between Figures 7 and 13. We feel that this finding is a result on its own and would not appear as evident to a new reader. Even if *a posteriori*, Figures 7 and 13 appears as redondant, we feel that, *a priori,* a new reader would ask to see the seasonality of the fluxes we are discussing, particularly during the peak period. Therefore we feel that it is justified to keep those 2 figures unchanged.**
**However, we will stress the little difference between both figures and what it means in the Discussion Section 4.3, p.16/l.18 : « Nonetheless, our conclusions are also valid on annual average since the drivers of nitrate and phytoplankton biomass in offshore boxes are similar on spring and annual average (see Fig. 7 and 13). »**

6) Section 3.2.1, line 32: What does "vertical velocities are pointing downward" mean in this context?

**This sentence will be modified as follows : « The vertical nitrate supply by advection off the northern and southern Saharan Bank is particularly weak (inward nitrate transport despite averaged outward velocities due to episodic inward events) in comparison to vertical diffusion. »**

7) Section 3.2.2, lines 20-25: Is there any other model or measurement data also showing the strong diffusion in this region? It would be nice to compare these results to published data or at least mention how reliable these results are.

**Indeed, the strong vertical mixing in this region is found in the global climatology of mixed-layer depth (MLD) of de Boyer Montégut et al. (2004), as mentioned in the Discussion Section 4.3.**

8) Section 4.1, lines 10-16 (page 14): Same as comment 7. Is there any model or measurement data to compare these results to?

**To the authors knowledge, no data or model results are available to validate our findings. These are actually innovative model results that would require further observational studies, as it will be stressed at the end of this Discussion Section 4.1., p.14/l.14 : « In the NW African region, coastal topography effects and alongshore geostrophic flow (related to large scale circulation patterns) may noticeably influence the convergence/divergence of coastal water masses. They would modulate the coastal divergence driven by the Ekman transport, i.e. the response of coastal upwelling to the wind forcing. This modeling work stresses processes that are yet difficult to study with observations due to their scarcity. Therefore, this work strongly advocates for dedicated observational studies. »**

9) Section 4.3, line 18 (page 17): Are the results of intrusion of nutrient-rich waters in the Senegalo-Mauritanian region in agreement with Lazaro et al., 2005 and Mittelstaedt, 1991? If so, it will be good to specify it: "originating from the Guinean upwelling. This is in agreement with previous finding of Lazaro et al., 2005 and Mittelstaedt, 1991." How do the results of Lazaro et al., 2005 and Mittelstaedt, 1991 compare to those of this paper?

**This sentence will be clarified as follows : « Nevertheless, our model results indicate that the nutrient input is not only from the coastal region. Indeed, we identified a significant impact of transient southern intrusions of nutrient-rich waters in the Senegalo-Mauritanian region likely originating from the Guinean upwelling due to the presence of the Guinea Dome, a large scale cyclonic feature centered south of the Cape Verde archipelago (Lázaro et al., 2005; Mittelstaedt, 1991). »**

10) Figure 14: Could these model parameters be compared to measurements, other model results or a theoretical average?

**The strength and seasonality of surface horizontal currents is validated against a global drifter-derived climatology of surface currents (see Fig. 1). Concerning vertical velocities which we mainly attribute to the coastal upwelling, they fall in the range of observed (Benitez-Barrios et al., 2011) or modeled values (Mason et al., 2011) off Morocco (so north of our study region). To the authors knowledge, observational data are missing in the study region to validate our estimations. This will be mentioned in the Results Section, p.7/l.11 : « Observational data are known to be scarce in our study region. However vertical velocities fall in the range of the few studies that has been published with observed (Benitez-Barrios et al., 2011) or**

**modeled values (Mason et al., (2011) focusing on northern Morocco. »**

11) Section 3.2.2 starts with "Annual mean PP", but it should be "Spring average PP...", I think.
**Indeed, this will be modified.**

12) Section "Discussion", line 11: This comment relates to point 5). Because it is not really mentioned before, I would shortly indicate, at the end of this paragraph, why spring processes are analyzed.
**See response to comment #5.**

13) Section 4.1, line 22 (page 13): "Therefore it gives us confidence..." how much confidence? Some, good, very good...? I would indicate it here.
**No data for quantitative validation is available but we can infer qualitatively that the model represents well the surface circulation of water masses in the study domain. This sentence will be modified as follows : « The simulated spatial and temporal variability of surface circulation are in good agreement with the satellite-tracked drifters (see Section 2.2), so the model can be used to infer the factors responsible for the sensitivity of coastal upwelling to the wind forcing. »**

14) Section 4.1, line 23 (page 13): "For this latter purpose, we further ..." This sentence does not really explain in what way the analysis of seasonal cycles at the edge of the coastal boxes will add confidence to the model. Especially because these sensitivity results are not compared to observations or theoretical behavior. Please be more specific about the importance of this analysis.
**We think that the correction made on the previous sentence (see previous comment) makes it clear that the analysis presented in Figure 14 allows to « infer the factors responsible for the sensitivity of coastal upwelling to the wind forcing. » (i.e. the « latter purpose »).**

15) Section 4.2, lines 33 (Page 14) and line 1 (page 15): Are these new results or do they include those of Barton et al., 2004, García-Muñoz et al., 2004 and Karakas et al., 2006? Please specify which part of these results are new or used to confirm previous ones.
**To clarify this point, we modified the sentence from :**
**« The combined effect of this local growth and the high filament activity around Cape Boujdour (Barton et al., 2004; García-Muñoz et al., 2004; Karakaş et al., 2006) results in an offshore transport of phytoplankton-rich and nitrate-depleted water masses. »**

**to :**

**« The combined effect of this local growth and the high filament activity around Cape Boujdour (the latter point being documented in Barton et al., 2004; García-Muñoz et al., 2004; Karakaş et al., 2006) results in an offshore transport of phytoplankton-rich and nitrate-depleted water masses. »**

16) Section 4.2, line 15 (page 15): "Zooplankton excretion..." What is the explanation of the zooplankton excretion being more important for regenerated production in areas with lateral transport?

**We actually mean that the zooplankton excretion « participates to enhance regenerated production in areas where the lateral input of plankton biomass is elevated » because zooplankton biomass, and then zooplankton excretion, is in this case high where the phytoplankton biomass is high. Accordingly, we propose to modify the end of the paragraph from p.15/l.9 :**
**« Usually the regenerated production relies on high residence time favourable to efficient recycling (see Fig. 15a). However, the water masses residence time in the south Saharan Bank and off Cape Blanc is low and can not explain the high level of regenerated production. In this region the regenerated production is rather due to the remineralization of organic matter supply and zooplankton excretion (Fig. 5a). Note that the meridional variability of secondary production (grazing rate) follows that of PP (not shown) suggesting a bottom-up control of the phytoplankton biomass rather than a top-down control by zooplankton grazers. Zooplankton biomass and excretion activity are then enhanced when the plankton biomass is elevated, which is especially the case in the South Saharan Bank and off Cape Blanc. »**

17) Section 4.3, lines 17-19 (page 15): "This supports the idea that, ..." Is this sentence in agreement or in opposition to the previous hypothesis? What do other authors say about lateral transport of organic matter in this area?

**This sentence is in agreement with the previous hypothesis which is that regenerated production is controlled by lateral inputs of organic matter (either living or dead) rather than by residence times. Indeed, in this area, the regenerated production is precisely low in spite of high residence times. Accordingly, we propose to modify the first sentences of the paragraph from p.15/l.16 :**
**« In the Senegalo-Mauritanian region, only moderate regenerated production is found year round although residence time is relatively high with respect to the southern Saharan Bank and Cape Blanc areas. This supports the idea that in the southern Saharan Bank, Cape Blanc and Senegalo-Mauritanian regions, regenerated production is rather driven by the amounts of organic matter supplies through lateral boundaries than by high residence time. »**

18) Section 4.3, lines 14-19 (page 16): "For this purpose, we focus...", this sentence should be briefly mentioned before as mentioned in comment 12, and here, it should be improve. Please clarify why this specific setting (maximal coastal upwelling at the Saharan Bank and maximum phytoplankton extension off Cape Blanc) is a good scenario to be tested with the model.

**As indicated to answer to comment 12, the choice of the spring season for the analysis of offshore boxes will be motivated in the introduction, and at the beginning of the Results Section 3.2.1. Moreover, the sentence concerned by this comment will be improved :**
**« For this purpose, we focus on the offshore region when the maximum chlorophyll extension is found (i.e. the spring period,**

Lathuiliere et al., 2008). In spring, maximum coastal upwelling is found off the Saharan Bank (Fig. 2c). Following the hypotheses of Lathuiliere et al. (2008), this should traduce in a maximum of phytoplankton biomass extension at the Saharan Bank latitudes. Instead, the phytoplankton biomass extension is found maximum off Cape Blanc and in the Senegalo-Mauritanian region, as attested by the meridional variation of phytoplankton biomass in the offshore region (Fig. 10). »

*Minor comments/Technical corrections*
**We agree with most of the minor comments made by the referee, and we wish to thank his/her for his/her great effort. We only answer to a few comments either to justify a disagreement, precise our thinking, or indicate an important correction that will be made to the manuscript.**

1) Section"Abstract" line 29: Did you mean "lateral advection transports coastal nutrients..."? **OK**

2) Section "Introduction" line 2: "... and seasonally variable one...". I would remove "one". **OK**

3) Section "Introduction" line 29: There is a missing comma, "Here, we ...", **OK**

4) Subsection "Model validation" line 22: Are AVHRR initials? If so, they probably should be written in full, here. **OK**

5) Subsection "Model validation" line 25: To ease the understanding of Figure 1, I would replace the titles "SeaWiFS/Drifters SVP" and "ROMS" by "Observations" and "Model", respectively. I could not figure it out what SVP means. Then, add in caption "from observation given by SeaWiFS satellite data... Same for ROM-PISCES model in..."
**We prefer to mention the name of the datasets shown on the validation Figure 1 as it does not requires more space and provide more informations to the reader. Nonetheless, we will change the titles to « SeaWiFS/Drifters GDP observations » and « ROMS model ». The GDP (Global Drifter Program) acronym will be added in the figure caption.**

6) Subsection "Model validation" line 6 and 7 (page 5): I would remove "see" from both "see Fig. 1" in this paragraph. I would also move the second fig. reference, line 7, to the end of the sentence: "both in the model and in the data during summer (Fig. 1)". This is, because the NECC is not entirely shown in the figure and creates some confusion if the figure reference is placed after mentioning the NECC. **OK**

7a) Subsection "Model validation" line 14 and Figure 1: If maximum values of chlorophyll in both satellite and model data go up to 10 mgChl m$^{-3}$, why stop at 5 mgChl m$^{-3}$ in the color bar of Fig. 1?
**We chose to stop at 5 mgChl m$^{-3}$ the color bar of Fig. 1 because there is increased overestimation of the satellite data with increasing**

**chlorophyll concentrations (Gregg and Casey, 2004).**

7b) Figure 1: At first I could not see the 5 coastal areas. Maybe by coloring the lines in white, instead of black, the boxes will be more visible. In the caption of Figure 1, "The ten boxes" could also be replaced by "The 5 coastal and 5 offshore boxes", so it is even more clear that the 5 coastal boxes are shown in the figure. **OK**

7c) Missing figure: Please remove the sentences "Main surface currents and deep water masses over … South Atlantic Central Water" or add the missing plot and describe it in the main text, but only if it adds relevant information to the article. **OK**

8a) Section 3.1, title: To simplify, why not just call it "Meridional variability in the coastal region"? **OK**

8b) Section 3.1, title:  Maybe add "Annual average of the meridional variability in the coastal region"
**We will change the title of Section 3.1 to « Meridional variability in the coastal region »**

9) Section 3.1.1, line 7:  "on the edge of coastal boxes" gives a vague description of what we are actually looking at in the figure.  Maybe replace by "on each edge of the coastal boxes (i.e. North, South, West and bottom)". **OK**

10a) Figure 2: Are the "BOX" and the black lines needed for figure 2a and 2b? I think they do not add much information, unlike for Fig.  2c, 2d, 2e, and 2f. These plots (Fig. 2a and 2b) alone clearly represent the entire box. **OK**

10b) Figures 2-3: These figures are good, but they are a bit difficult to get at first (especially Fig. 2, since it is the first one of the series). To make Fig. 2 easier to understand, I would separate the main title "Wind and currents" to "Wind (m s$^{-1}$)" and "Velocities (m s$^{-1}$)". This will make it clear that this figure is showing 2 different things. I would write "Wind (m s$^{-1}$)" on top of Figure 2a and 2b and "Velocities (m s$^{-1}$)" on top of Figures 2b, 2c, 2d, 2e and 2f, as done for Fig.  3.  Please use velocities instead of currents, because the word current is not as often used as velocity in the main text. Please also change the title of figure 2c to "c-Bottom (upwelling)" or "c-Bottom (vertical velocities)". The chosen title,  "c-Bottom velocity" is misleading,  as it suggests horizontal bottom velocities. **OK**

10c) Please check the sub-numbering of the figures, some do not follow a logic order. **OK**

11) Section 3.1.2, line 30:  Please rephrase "…  of PP in the boxes north and south of Cape Blanc are simulated, respectively." **OK**

12) Section 3.1.3, lines 26-30:  This sentence is difficult to follow, maybe rephrase to "… the total rate of change of nitrate concentration and phytoplankton biomass in each coastal box are presented…" **OK**

13) Section 3.1.3, line 8: replace "is not anymore" by "is no longer" **OK**

14) Section 3.1.3, line 13: rephrase "from the northern Saharan Bank and the Senegal-Mauritanian region, respectively." **OK**

15) Section 3.1.3, line 31 (page 9): remove "tentative" **OK**

16a) Section 3.2, title: Same as comment 8: To simplify, change it to "Meridional variability in the offshore region". **OK**

16b) I think, here, the title of section 3.2 should also mention that the results are now only for spring: "Spring meridional variability in the offshore region". For me this was not clear until the discussion.
**We agree and will change the title of Section 3.2 to « Spring meridional variability in the offshore region ».**

17) Section 3.2.2, line 27: I would remove "finally". **OK**

18) Section 4, line 10: Please replace "we will seek to explicit" by "we explain" **OK**

19a) Section 4.3, line 10 (page 17): Space missing, please correct "that filaments" **OK**

19b) Section 4.3, line 14(page 17): Please replace "hypothesis" by "processes". The points mentioned before are not hypotheses. **OK**

20) Figure 8: Same comment as 10a for Figure 2 regarding the black lines and the BOX. **OK**

21) Figure 11a and 11b are missing.
**This should be « a- Bottom advection » and « b- Bottom diffusion », so this will be modified.**

22) Figure 14: Why Fig 14c is suddenly represented with dashed lines? What do the plain and dashed lines represent in Fig. 14d? **OK**

23) Please check the usage of acronyms, especially in the legend of the figures. I would always use the full subregion names or always the acronyms. Personally, I would avoid the acronyms (also in the text).
**We agree and will always use the full subregion names.**

24) Section 4.3, line 8: Please correct "2ZOLTLathuiliere2008" **OK**

25) I would replace most "see Fig." by "Fig.", except maybe the one in line 24, page 4. **OK**

_Further suggestions_
**We agree with all these further suggestions made by the referee, and we wish to thank her again for his/her great effort.**

1) Section 4.1, line 15-16 (page 13): Rephrase "It appears that coastal upwelling" by "However, coastal upwelling". This makes it more clear that

this information comes from the literature and not the model results. **OK**

2) Section 4.1, line 20 (page 13): Replace "The simulated" by "Our simulated"
**OK**

3) Section 4.2, line 4: "for 50 % of new production", please indicate if it is a model result or a general statement. Can this be seen in one of the figures? If yes, please indicate in which one.
**This is shown in figure 6.**

4) Section 4.2, lines 6-8:  Would it be possible to compare this with observations of mixed layer depth (MLD) off Cape Blanc?  If the MLD is large there, it will give some extra validation for this result.
**At the coast off Cape Blanc, there is phytoplankton subduction by advection and not by diffusion, so this feature is not related to the vertical mixing.**

5) Section 4.2, lines 9-10: I would replace "Usually" by "In general", and in the following sentence, I would add "However, in our results, the water ...." **OK**

6) Section 4.2, line 35 (page 15): The word "provide" is maybe not the best word, here. I suggest "lead to", instead. **OK**

7) Section 4.2,  lines 1-2 (page 16):  To ease the reading,  I would briefly remention why this is the case, instead of referring to the previous section. Section 4.1 mentions several points and it is not exactly clear to which one this sentence refers to.  I would also remove "(see above)", it is not clear to what it is referring to. **OK**

8) Section 4.3, line 27:  Please add "In our results, the vertical mixing..." and replace "Indeed" by "This is even more visible in the results for spring, where the mixed layer...". **OK**

9) Section 4.3, lines 33-34:  To better differentiate between the new results and published data, please rephrase the sentence as follows:  " In fact, the vertical mixing, as previously suggested by Huntsman and Barber 1977, is also responsible for ..." **OK**

10) Section 4.3, line 1: Maybe replace "participate to" by "partly". **OK**

11) Section 4.3, line 7:  To help the reader, it would be nice to briefly mention what is the hypothesis of Lathuilère et al., 2008. **OK**

12) Section 4.3,  lines 8-9:  To better differentiate between the new results and published data, please add " In our results, the advection by ..." and "...off Cape Blanc, in agreement with Kostianoy and Zatspein, 1996. **OK**

13) Section 4.3, line 10: To better differentiate between the new results and published data, please add "may enhanced cross shelf transport, as also shown by satellite data in Lathuilère et al., 2008". **OK**

---

## Author Comment (AC3) · 30 Sep 2016

We wish to thank referee #3 for his/her detailed analysis and his/her thoughtful comments, which will improve the quality of this manuscript. Here, you will find a detailed reply to each comments :

Response to Referee#3's Comments

*General comments*
Title could be shortened to:  What drives the spatial variability of primary productivity and matter fluxes in the North-West African upwelling system?  A modelling approach.
**We agree and will shorten the title as suggested by the referee.**

Throughout the manuscript 'explicit' is used as a verb. It is not a verb, it can be used as a noun (eg. The explicitness of the data allow us to draw some very solid conclusions) but usually an adverb (The data allow us to explicitly show that...) or adjectve (e.g. The data is explicit, it shows that...).
**The term « explicit » will be replaced by an adequate verb each time it was badly used in the manuscript.**

It seems a bit unclear as to why the spring means are used for the offshore domains and annual means for the coastal domains. It is mentioned only later in the discussion, but it should be clearer sooner. Why not show the spring mean for coastal and offshore domains (surely using the annual mean for the coastal domain masks the seasonal signal and is therefore unrealistic?). Does it make sense to link the coastal and offshore domains in terms of the offshore fluxes for example if you are looking at averages for different periods?
**This comment is related to comments of the 2 other referees who asked for introducing the choice of the spring period earlier in the paper.**
**The main justification is that observed offshore extension of Chl-a do present a marked seasonal variability with a peak in boreal spring. Therefore, focusing only on annual averages would have raised questions about the significance of our results during the time period that sees most of the offshore export. Choice has thus been made to show annual average but also the spring period.**
**Interestingly, there is very little change in the repartition of the fluxes driving nitrate and phytoplankton concentrations in spring compared to annual average, as shown by the little difference between Figures 7 and 13. We feel that this finding is a result on its own and would not appear as "evident" to a new reader. Even if *a posteriori*, Figures 7 and 13 appears as redundant, we feel that, *a priori,* a new reader would ask to see the seasonality of the fluxes we are discussing, particularly during the peak period. Therefore we feel that it is justified to keep those 2 figures unchanged.**
**However, we will stress the little difference between both figures and what it means in the Discussion Section 4.3, p.16/l.18 : « Nonetheless, our conclusions are also valid on annual average since the drivers of nitrate and phytoplankton biomass in offshore boxes are similar on spring and annual average (see Fig. 7 and 13). »**

The results section is very laborious and therefore difficult to read, especially sections 3.1.3 and 3.2.3 (the percentages in paretheses are not necessary),

and could be shortened.
**According to the referee's suggestion (shared with referee #1), numbers in sections 3.1.3 and 3.2.3 will be removed from the text. The Results Section will also be clarified to higlight the major features of the study region shown in the figures.**

**We agree with most of the minor comments made by the referee, and we wish to thank him/her for his/her great effort. We only answer to a few comments either to answer a question or indicate an important correction that will be made to the manuscript.**

*Specific comments*
page 2, line 4: 'of coastal topography are' should be 'of coastal topography is'
**OK**

page 2, line 5: '...and then the response of nutrient upwelling to wind forcings'. This is unclear. Are you saying that the large scale circulation pattern impacts the wind driven upwelling of nutrients?
**Large scale circulation patterns actually affect the response to the wind forcing of the vertical velocities at 100m depth (which we attribute to coastal upwelling), and so the vertical nutrient inputs in the euphotic layer.**

*Introduction*
Page 3, line 17: 'in regards of environmental forcings' should be 'with regard to environmental forcings' **OK**

Page 4, line 2: 'To this end, comparative box analysis...' should be 'To this end, a comparative box analysis...' or 'To this end, comparative box analyses......have been conducted'. **OK**

Page 4, line 3, 'Those subregions' should be 'The subregions' **OK**

Page 4, line 4,' in regards of' should be 'with regard to' **OK**

Page 4, Line 14: 'explicit' cannot be used as a verb, try 'identify', **OK**

Last two paragraphs of Introduction are laborious and could be more succinctly summarised.
**According to the referee's comment, the last two paragraphs will be modified as follows :**
**« First, we present the model configuration and a validation of near-surface circulation and surface chlorophyll biomass using in-situ and satellite data (Section 2.1 and 2.2, respectively). Then, we describe the meridional variability of wind forcings, ocean response and primary productivity as simulated by the model in the different coastal (Section 3.1) and offshore boxes (Section 3.2), on annual mean and also during spring (seasonal maximum of the chlorophyll offshore extension as shown in Lathuilière et al, 2008). Each section is split in three parts which describe the meridional variability of (i) the wind**

**forcings, current velocity and nitrate fluxes, (ii) the primary production (PP), phytoplankton biomass and phytoplankton fluxes, and (iii) the sources and sinks of nitrate concentration and phytoplankton biomass. Finally, we discuss in Section 4 (i) the sensitivity of coastal upwelling to the wind forcing along the NW African coast, (ii) the meridional variability of coastal phytoplankton biomass and PP (new and regenerated production) in relation with matter transfers and (iii) the meridional variability of the offshore extension of coastal chlorophyll off NW Africa. »**

Last two sentences of Introduction seem out of place.
**Indeed, these two sentences will be removed.**

*Methods*
Page 4,line 26: unbalanced parenthesis **OK**

Page 5, line 23: 'thinner' is ambiguous in this context, 'narrower' is clearer **OK**

Page 6, line 3: 'The upwelling filaments off Cape Ghir and Cape Boujdour are responsible for strong seaward deflections of the coastal current.' I wouldn't necessarily say that the filaments are responsible for the seaward deflection – they are 'connected', both associated with the same initial mechanism (perhaps wind/topography) and then they probably enhance one another. **OK**

Page 6, Line 2: 'explicit' cannot be used as a verb, try 'identify' **OK**

Page 6, line 2-4: '..........the meridional variability of primary productivity off the NW African coast, we carried out a box analysis focusing on nitrate (the main limiting nutrient) and phytoplankton carbon budgets (12–27°N, see Fig. 1)', rather say:'..........the meridional variability of primary productivity off the NW African coast between 12–27°N, we carried out a box analysis focusing on nitrate (the main limiting nutrient) and phytoplankton carbon budgets' **OK**

Page 6, line 7: '...was split ito five latitudinal bands.' rather: '...was split into five latitudinal bands (see Fig. 1).' **OK**

Page 6, line 12: remove 'On the opposite,', start with 'In the southernmost...'
**OK**

Page 6, line 17: what do you mean by 'globally'? It usually refers to something involving the whole globe/world.
**The term 'globally' will be removed since it is not essential.**

Page 6, line 20: 'In like manner...' rather 'Similarly..' **OK**

*Results*
Page 8, line 6: 'Wind curl shows a clear maximum off Cape Blanc but a weak meridional variability'. This sentence is not clear. Do you mean that the meridional variability in wind stress curl is weak or do you mean that the alongshore variability of meridional wind stress is weak?
**We mean that the meridional variability in wind stress curl is weak. This will be clarified (see General Comment #4).**

Figure 2 c : this is labelled as upwelling intensity (vertical velocity at the bottom). For upwelling intensity, it would be better to use vertical velocity at the base of the Ekman layer (your 100 m depth of the boxes is probably too deep?).

**In this paper, we are interested in the primary productivity. In consequence, we designed boxes extending vertically from the free surface down to 100m depth to encompass the euphotic layer where light is available for phytoplankton photosynthesis and primary production. We then consider the upwelling-induced nutrient flux at the base of the boxes (100m depth) since the nutrients that enter this layer are susceptible to be consumed for primary production.**

Page 8, line 24: remove 'inversely' **OK**

Page 8, line 32: '...associated to...' should be '...associated with...' **OK**

Page 9, line 11: 'does not translate in...' should be 'does not translate into....' **OK**

Page 9, line 14: 'Noteworthy, the phytoplankton biomass is found maximum off Cape Blanc and the South Saharan Bank contrasting with minimum upwelling-induced nitrate supplies (Fig. 3a).
rather:
'It is noteworthy that maximum phytoplankton biomass is found off Cape Blanc and the South Saharan Bank despite the fact that upwellling-induced nitrate supplies are at a minimum at those locations (Fig. 3a).' **OK**

Section 3.1.3: laborious
**According to the referee's suggestion (shared with referee #1), numbers in Sections 3.1.3 will be removed from the text. The Results Section will be generally clarified to higlight the major features of the study region shown in the figures.**

page 10, line 6: 'sinks' should be 'sink' **OK**

page 11, line 12: 'enlightened' is very archaic in this context. Replacing it with 'euphotic zone' would be better. **OK**

Page 11, line 14: 'explicit' cannot be used as a verb and whole sentence is unclear. **OK**

Page 11, line 17: 'Alternatively' since you're not offering an alternative to a previous statement, something like 'On the other hand' works better. **OK**

Page 11, line 22: 'Noteworthy, at the western...', change to 'It is noteworthy that at western boundaries velocities are...' **OK**

Page 11, line 23: replace 'happen to be' with 'are' **OK**

Page 11, line 29: replace '..falls in the same order of magnitude than diffusion...' with '..is the same order of magnitude as diffusion...' **OK**

Page 11, line 31: replace 'Noteworthy...' with 'It is noteworthy that vertical nitrate supply...' **OK**

Page 12, line 10: In the text it states that Fig. 10 is annual mean, but the figure caption says Spring mean. **OK**

Page 12, line 18: replace '......, in less manner,....' with '..., less so,...' **OK**

Page 13, line 2: In the text it states that fig 12 shows the annual mean source and sink terms but the figure caption says it is the spring mean. **OK**

*Discussion*
Page 14, line 9: 'in relation with the...', should be 'in relation to the...' **OK**

Page 14, line 10: 'Finally, we will seek to explicit...' should be 'Finally, we will seek to identify...' **OK**

Page 14, line 13: 'In our simulation, the meridional variability of coastal upwelling is not correlated to the local variability of wind-driven Ekman transport and Ekman pumping. This result questions the estimation of vertical velocities based on local wind forcing that were commonly used in EBUS'. **OK**

- two points on this statement: 'were' should be 'are' **OK**

The estimation of upwelling using alongshore wind stress is for vertical velcoities at the base of the Ekman layer.  Your level of 100m, or the bottom in places shallower than 100 m, may be too deep.
**We agree and the sentence will be modified as follows : « In our simulation, the meridional variability of vertical velocities at 100m depth (which roughly corresponds roughly to the euphotic layer) is not correlated to that of upwelling-favourable winds and Ekman pumping. This result questions the estimation of upwelling-induced nutrient inputs in the euphotic layer based on the wind-driven Ekman transport and the nutrient concentrations in upwelling source waters, a method commonly used in EBUS (Gruber et al., 2011; Messié et al., 2009; Messié and Chavez, 2014). »**

Page 14, line 16:  You state that the large scale transport could be a factor explaining the mismatch in upwelling intensity and Ekman transport.  With the model output you can calculate it directly to verify your statement.
**The mismatch between Ekman transport and upwelling intensity off Cape Blanc implies that other mechanisms than the Ekman transport play against coastal upwelling to create downward vertical velocities. These can be internal waves, mesoscale processes (like fronts and eddies) or the convergence of water masses. We clearly show that downward vertical velocities in the coastal box off Cape Blanc from May to July co-occur with alongshore velocities at both the northerm and southern boundaries directed inward, which demonstrates that the convergence of water masses is the most plausible explanation for this mismatch.**

Page 14,line 22: '...explicit...', can't be used as a verb. You could use 'identify' **OK**

Page 14, line 23 and figure 14: you use the bottom velocity to assess the sensitivity of coastal upwelling to wind forcing. You should rather use vertical velocity at the base of the Ekman layer.
**The point here is to investigate the sensitivity to wind forcing of the vertical velocities at 100m depth (attributed to the coastal upwelling) that participate to the vertical nutrient fluxes in the euphotic layer where nutrients are consumed for primary production (see our response to the comment in Section Results « Figure 2 c »).**

Page 14, line 26: 'lead' should be 'leads' **OK**

Page 15, line 18: instead of 'Albeit' use 'Although'. **OK**

Page 16, line 31: sentence starting with 'This points a gap...' is confusing **OK**

Page 17, line 13: 'the coast, the wind stress curl...' should be 'the coast, (ii) the wind stress curl...' **OK**

Page 17, line 14: 'hypothesis' should be hypotheses' **OK**

Page 17, line 15: 'explicit' should be 'identify' **OK**

Page 17, line 20: 'associated to' should be 'associated with' **OK**

Page 17, line 22: 'Our results indicate downward and upward wind-induced Ekman pumping of respectively north and south of Cape Blanc' should be 'Our results indicate downward and upward wind-induced Ekman pumping north and south of Cape Blanc respectively' **OK**

Page 18, line 1: 'participate' should be 'help' **OK**

Page 18, line 8: '2ZOTLathuliere2008' – a latex referencing bug? **OK**

Page 18, line 10: 'thatfilaments' should be 'filaments' **OK**

*Conclusion*
Page 18, line 21: 'of the primary production spatial distribution in' should be 'of the spatial distribution of primary production' **OK**

Page 18, line 25: ' production in' should be 'production with' **OK**

Page 18, line 31: 'excepted' should be 'except' **OK**

*Figures*
Figure 1: include the box labels that you use in the text and in other figures **OK**

Figures 1-4: in some you include just the abbreviations of the box areas, in others you have the full name. When you don't have the full names in the legend, you could include them in the caption **OK**

Figure 6: label x-axis with latitude as well, or at least show where north is **OK**

Figure 10 and 12: the captions dont agree with the text (Annual vs. Spring mean) **OK**

Figure 14:  it is not clear how these averages are calculated (in the caption or in the text).  In the caption you state:  'within and at the boundaries of coastal boxes'.  Is it an average of meridional wind, bottom velocity, cross-shore velocity and alongshore velocity within the entire coastal strip?  If so, does this make sense, given that your interest is the meridinal variability of primary productivity

**In the text, it will be modified as : « For this latter purpose, we further analyze the seasonal cycles of meridional wind versus vertical and horizontal velocities averaged within and over each edge of the coastal boxes (i.e. North, South, West and bottom), respectively (Fig. 14). »**

**The caption will also be modified as follows : « Figure 14: Seasonal climatology of (a) wind intensity (negative is upwelling-favourable, m s$^{-1}$), (b) bottom vertical velocity (m s$^{-1}$), (c) zonal velocities (m s$^{-1}$) and (d) meridional velocities (m s$^{-1}$) averaged within and over each edge of the coastal boxes (i.e. North, South, West and bottom ; defined positive inward, so vertically upward), respectively. Each color corresponds to a box (see legends in Fig. 2). In (d), a solid (dashed) line represents a velocity at a northern (southern) edge of a box, respectively. »**